# Negative Regulation of the Innate Immune Response through Proteasomal Degradation and Deubiquitination

**DOI:** 10.3390/v13040584

**Published:** 2021-03-30

**Authors:** Valentina Budroni, Gijs A. Versteeg

**Affiliations:** Max Perutz Labs, Department of Microbiology, Immunobiology, and Genetics, University of Vienna, Vienna Biocenter (VBC), 1030 Vienna, Austria; valentina.budroni@univie.ac.at

**Keywords:** ubiquitin, deubiquitinase, E3 ligase, innate immune system, proteasome, cytokine

## Abstract

The rapid and dynamic activation of the innate immune system is achieved through complex signaling networks regulated by post-translational modifications modulating the subcellular localization, activity, and abundance of signaling molecules. Many constitutively expressed signaling molecules are present in the cell in inactive forms, and become functionally activated once they are modified with ubiquitin, and, in turn, inactivated by removal of the same post-translational mark. Moreover, upon infection resolution a rapid remodeling of the proteome needs to occur, ensuring the removal of induced response proteins to prevent hyperactivation. This review discusses the current knowledge on the negative regulation of innate immune signaling pathways by deubiquitinating enzymes, and through degradative ubiquitination. It focusses on spatiotemporal regulation of deubiquitinase and E3 ligase activities, mechanisms for re-establishing proteostasis, and degradation through immune-specific feedback mechanisms vs. general protein quality control pathways.

## 1. Regulation of Innate Immune Signaling by Ubiquitination

The innate immune system is the host’s front line of defense against invading pathogens such as viruses and bacteria. Intra- and extracellular pattern recognition receptors (PRR) such as Toll-Like Receptors (TLR), Retinoic Acid-Inducible Gene-I (RIG-I)-like Receptors (RLR), Nucleotide-binding Oligomerization Domain (NOD)-Like Receptors (NLR), and DNA sensors recognize pathogen-associated molecular patterns (PAMP) of microbes [1,2,3]. Subsequently, they activate several signaling cascades, culminating in the activation of transcription factors including Nuclear Factor κB (NFκB), Activator Protein 1 (AP1), and members of the Interferon (IFN) Regulatory Factor (IRF) family, which induce the expression of proinflammatory cytokines, chemokines, and type I interferons [1,2,3]. The latter, in turn, activate downstream signaling pathways to induce the expression of hundreds of antiviral interferon stimulated genes (ISG), which are potent antiviral restriction factors [1,2,3].

The rapid and dynamic activation of the immune system is achieved through complex signaling networks regulated by post-translational modifications modulating the subcellular localization, activity, and abundance of signaling molecules [4]. Ubiquitination has emerged as an important mechanism to activate and repress the innate and adaptive immune responses.

Many constitutively expressed signaling molecules are present in the cell in inactive forms, become functionally activated once they are modified with ubiquitin, and in turn inactivated by removal of the same mark [3,5,6]. Moreover, upon infection resolution, a rapid remodeling of the proteome needs to occur, ensuring the removal of a plethora of transcriptionally induced response proteins (e.g., transcription factors, antiviral factors, effector proteins [7,8,9,10,11,12]) to restore homeostasis and prevent hyperactivation, and chronic inflammation [3,5,6].

For short-lived cells, such as neutrophils, this may be irrelevant, as their own programmed death ensures removal of activated mediators. Moreover, rapidly dividing cells may simply dilute out part of their activated proteasome once de novo response protein synthesis ceases. In contrast, slowly-dividing or post-mitotic cell types must actively remove these response proteins. To understand meaningful biology, in this context it is important to distinguish specific cellular mechanisms that degrade, e.g., activated response proteins, from non-specific cellular protein quality control pathways that degraded, e.g., misfolded proteins.

Immune-specific degradation pathways must be controlled by either feedback or feed-forward mechanisms, ensuring correct spatiotemporal, and specific protein degradation. Alternatively, some immune response proteins may be inherently unstable because they harbor a degron, and their steady-state equilibrium is determined by the rate of stimulus-dependent de novo synthesis on the one hand, and its degradation on the other.

In this review, we discuss the current knowledge on the negative regulation of the innate immune signaling pathways by deubiquitinating enzymes (DUB), and by degradative ubiquitination, emphasizing the following aspects: (i) Spatiotemporal regulation of DUB and E3 ligase activities: What determines their abundance, cellular localization and timing of their activity in order to resolve immune signaling? (ii) Re-establishment of proteostasis: What happens to newly synthesized antiviral response-molecules after clearance of infection? (iii) Negative regulation of immune signaling by protein degradation: which are dedicated, immune-specific feedback mechanisms, and which may be part of general protein quality control pathways?

## 2. Ubiquitin as a Versatile Signal to Regulate the Immune Response

Ubiquitin is a small, 8.5 kDa protein which is covalently attached via its C-terminal glycine to predominantly lysine side chains of target proteins through a sequential cascade of ubiquitin-activating (E1), ubiquitin-conjugating (E2), and ubiquitin-ligating (E3) enzymes, the so-called ‘writers’ of the ‘ubiquitin code’ [13,14]. Ubiquitin itself contains seven lysine residues (K6, K11, K27, K29, K33, K48, K63) which, together with the N-terminal methionine, enable the modification of target proteins with eight different poly-ubiquitin chains (conjugation of ubiquitin molecules via the same lysine residue), heterotypic ubiquitin chains (conjugation via different linkage types), branched chains, or mono-ubiquitination [15].

Depending on the poly-ubiquitin linkage type, these chains trigger different cellular outcomes: K48-linked polyubiquitin chains are the most abundant ubiquitin chain type in cells and represent the predominant signal for proteasomal degradation [16]. Instead, M1 and K63-linked polyubiquitin chains have mostly proteasome-independent functions, with important activating roles in immune signaling [3,14]. Other ubiquitin chain types (K6, K11, K27, K29, K33) are far less characterized, however their involvement in antiviral innate immune signaling becomes increasingly evident, having both activating and degradative functions [16,17].

While the human genome encodes only two E1 enzymes for ubiquitin and ~40 E2 enzymes, more than 600 genes encoding E3 ligases have been annotated [18,19,20]. Since E3 ligases are the most numerous group, they are thought to determine substrate specificity, whereas chain topology is determined by E2 and E3 enzymes, depending on their class [21,22].

E3 ligases have been classified according to the architecture of their catalytic domain, and their mechanism of substrate recognition. The largest group of Really Interesting New Gene (RING) E3 ligases catalyze the transfer of ubiquitin directly from the E2 enzyme to the substrate [23]. Instead, the Homologous to E6-AP Carboxyl Terminus (HECT) E3 ligases form an intermediate (E3-Ub) before ubiquitin is transferred to the substrate [24]. The last group of RING-Between-RING (RBR) E3 ligases combines both mechanisms: its RING1-domain recognizes ubiquitin on the E2 enzyme, which is then transferred to its RING2-domain, and further to the substrate [19,25].

The readers of the resulting ‘ubiquitin code’ are proteins harboring distinct Ubiquitin-Binding-Domains (UBD). These UBDs represent structurally very distinct motifs that can recognize different ubiquitin linkages [26], which in turn is translated in biological output through conformational changes, and multimerization.

Finally, ubiquitination marks are erased by DUBs, which hydrolyze the covalent bond between ubiquitin residues or between ubiquitin and the target protein [27]. Compared to the number of E3 ligases, the number of DUBs is relatively small, with less than 100 predicted DUB sequences in the human genome, of which only ~80 are predicted to encode functional proteins [28].

DUBs are grouped into eight classes, according to their enzymatic cleavage preferences: (**i**) The Ubiquitin-Specific Proteases (USP/UBP), (**ii**) Ubiquitin C-terminal Hydrolases (UCH), (**iii**) Ovarian Tumor Proteases (OTU), (**iv**) Machado-Josephin Domain (MJD) DUBs, (**v**) Motif Interacting with Ub-containing Novel DUB (MINDY; cysteine proteases), (**vi**) c-Jun Activation domain-Binding protein-1 (JAB1)/Moloney leukemia virus-34 Proviral Integration (MOV34)/Mannose 6-Phosphate Receptor 1 (MPR1)/Proteasome-Associated PAD1 Homolog (PAD1) N-terminal (MPN) Domain-Associated Metallopeptidases (JAMM), (**vii**) Monocyte Chemotactic Protein-Induced Protein (MCPIP), and (**viii**) the recently discovered Zinc Finger with Ubiquitin Fold Modifier 1 (UFM1)-Specific Peptidase Domain Protein proteases (ZUFSP) [29,30].

Many E3 ligases and DUBs have been implemented in the regulation of immune signaling pathways by synthesizing or removing activating or degradative ubiquitin marks, thereby modulating the activity and stability of signaling molecules. Once the infection is cleared, the termination of the immune response is critical to prevent chronic inflammation and autoimmune diseases [31].

Besides coinhibitory molecules such as Cytotoxic T-lymphocyte-associated protein 4 (CTLA-4) and Programmed Cell Death 1 (PD-1), post-translational modifications (e.g., phosphorylation, ubiquitination) represent another important regulatory mechanism to regulate the activity and abundance of signaling molecules and transcriptionally upregulated effector molecules (e.g., transcription factors, antiviral proteins) [4,32]. In fact, several negative regulators of immune signaling have been found to be mutated in chronic and autoimmune diseases [33,34,35,36,37]. In this review, we focus on factors and mechanisms regulating ubiquitin-dependent innate immune responses, with a focus on negative regulation by DUBs, and proteasomal degradation.

## 3. Negative Regulation of RLR Signaling

RIG-I, Melanoma Differentiation-Associated Protein 5 (MDA5), and Laboratory of Genetics and Physiology 2 (LGP2) are part of the family of RIG-I like receptors (RLR) that recognize viral 5′ppp and double-stranded RNA in the cytosol of infected cells [38]. RIG-I and MDA5 are crucial for the innate antiviral immune response, as mice lacking RIG-I or MDA5 are highly susceptible to infection, and fail to produce type I IFNs and proinflammatory cytokines [38,39,40].

RLRs share a conserved DExD/H-box helicase core and a C-terminal domain (CTD), critical for RNA sensing. In addition, RIG-I and MDA5 share two N-terminal Caspase Activation and Recruitment Domains (CARD), required for the interaction with the mitochondrial antiviral signaling adapter (MAVS; also known as CARDIF, IPS-1, VISA) [41,42,43].

MAVS recruits Tumor Necrosis Factor (TNF) Receptor Associated Factor 3 (TRAF3), which activates the downstream kinases TRAF Family Member Associated Nuclear Factor κB (NFKB) Activator (TANK) Binding Kinase 1 (TBK1) and Inhibitor Of Nuclear Factor Kappa B Kinase Subunit Epsilon (IKKε; Figure 1) [44,45]. These, in turn, phosphorylate the transcription factors IRF3 and IRF7, which homo- and heterodimerize, translocate to the nucleus and promote the expression of type I IFNs [31,46]. In addition, MAVS signals to TRAF6, which recruits Transforming Growth Factor Beta 1 (TGF)-Beta Activated Kinase 1 (TAK1)/TGF-Beta Activated Kinase 1 Binding Protein 1 (TAB1)/TAB2/3 and Inhibitor of Nuclear Factor Kappa-B Kinase Subunit Alpha (IKKα)/Inhibitor of Nuclear Factor Kappa-B Kinase Subunit Beta (IKKβ)/NFκB Essential Modulator (NEMO), responsible for NFκB activation [47].

RLRs are constitutively expressed, and reside in the cytoplasm of uninfected cells in an auto-repressed, inactive state [48,49]. Viral RNA binding to RIG-I triggers its conformational change and oligomerization, promoted by ubiquitination by the E3 ligases Tripartite Motif Containing 25 (TRIM25) and Ring Finger Protein 135 (RNF135/RIPLET) with unanchored K63-linked, and covalently attached K63-linked ubiquitin chains in its CARD domains, forming a helical tetrameric “lock washer” structure [49,50,51,52,53,54,55]. RIG-I oligomers nucleate the formation of MAVS prion-like aggregates on the mitochondria, which are further stabilized by K63-linked ubiquitination by TRIM31 and potently activate TBK1/IKKε [56,57].

The mechanism of MDA5 activation is less well understood. Structural studies showed that MDA5 forms filaments along dsRNA, which activate and promote the formation of MAVS aggregates [58]. Both unanchored, as well as K63-linked ubiquitin chains synthesized by TRIM65 have been proposed to promote its oligomerization [59,60].

TRAF3 and TRAF6 are recruited to activated MAVS and subjected to ubiquitination with K63-linked chains. K63-linked ubiquitin chains serve as a scaffold for the recruitment of the downstream kinase complexes TBK1/IKKε, or TAK1/TAB1/TAB2/3 and IKKα/IKKβ/NEMO, respectively [44,45,47,61,62]. Activation of TBK1 and IKKɛ also strongly depends on ubiquitination with K63-linked ubiquitin chains [63,64,65,66].

A plethora of DUBs and E3 ligases have been reported to negatively regulate RIG-I signaling by removing activating ubiquitin chains from various signaling molecules, or catalyzing their ubiquitination with degradative ubiquitin chains, thereby attenuating immune signaling.

Well-studied immune proteins have often been reported to be post-translationally controlled by a multitude of regulators, which raises the question why so many enzymes would be engaged on single targets. For instance, at least six DUBs (Cylindromatosis (CYLD) [67,68], USP3 [69], USP21 [70], USP14 [71], USP25 [72], and USP27X [73]) have been proposed to counteract K63-linked ubiquitination of the RNA-sensor RIG-I, in addition to at least six E3 ligases (RNF125 [74], CBL [75], RNF122 [76], Stress Induced Phosphoprotein 1 (STIP1) Homology And U-Box Containing Protein 1 (STUB1) [77,78], Membrane Associated Ring-CH-Type Finger 5 (MARCH5) [79], and TRIM40 [80]), which reduce protein levels of RIG-I via the ubiquitin-proteasome system.

RIG-I expression is increased by IFN signaling and its protein levels and activity must be tightly regulated to ensure return to homeostasis and prevent hyperactivation of IFN and cytokine signaling [81,82]. However, this multitude of DUBs and E3 ligases may suggest a certain degree of redundancy, possibly accompanied by cell-type-specific roles of the distinct enzymes, and different spatio-temporal activities. Moreover, for many E3 ligases and DUBs their relevance in vivo has not been determined yet. In the following section we will discuss DUBs and E3 ligases involved in the negative regulation of RLR induced type I IFN production.

### 3.1. DUBs Inhibiting RLR Activation

Several members of the USP DUB family have been proposed to remove K63-linked ubiquitin chains from RIG-I and MDA5 to prevent exaggerated immune responses.

Firstly, CYLD was shown to remove K63-linked ubiquitin chains from RIG-I, TBK1, and IKKɛ under basal conditions, thereby inhibiting RLR signaling [67,68]. Infection of *Cyld^−/−^* Dendritic Cells (DC) and Mouse Embryonic Fibroblasts (MEF) with vesicular stomatitis virus (VSV) resulted in hyperactivation of TBK1 and IKKɛ, and consequently hyper-induction of type I IFNs due to increased levels of RIG-I ubiquitination [67,68]. Importantly, cellular CYLD protein levels were reduced in the presence of TNF and Sendai virus (SeV) infection, likely increasing RIG-I activation, and suggesting a feedback mechanism to lower the concentrations of this RIG-I DUB during activation [67].

Unexpectedly, *Cyld* ablation in mice decreased survival upon sub-lethal infection with VSV compared to their wild-type (WT) littermates and accordingly, *Cyld^−/−^* dendritic cells had enhanced susceptibility to VSV infection [83]. Importantly, this did not stem from reduced type I IFN production, but resulted from decreased Signal Transducer And Activator Of Transcription 1 (STAT1) phosphorylation and ISG expression, suggesting that CYLD might also regulate Interferon Alpha And Beta Receptor (IFNAR) signaling [83].

Similarly, USP21 was shown to interact with RIG-I both in uninfected and VSV- and SeV-infected cells and to remove K63-linked ubiquitin chains from the CARD domains of both RIG-I and MDA5 [70]. *Usp21^−/−^* mice were resistant to VSV infection, resulting from increased type I IFN production [70]. Accordingly, deficiency of *Usp21* in peritoneal macrophages and bone marrow derived dendritic cells (BMDC), but not bone marrow derived macrophages (BMDM), increased ubiquitination of RIG-I, IRF3 phosphorylation, and levels of IFNα and IFNβ in response to SeV infection, indicating that the activity of USP21 could have different roles in different cell-types [70].

The fact that USP21 interacts with RIG-I irrespective of infection may suggests that it plays predominantly a constitutive role in inactivation. Alternatively, immune activation could activate USP21 post-translationally, thereby providing an activated feedback. While more studies will be required to distinguish between these options, a related family member—USP3—was shown to interact with RIG-I specifically upon viral infection and to attenuate RLR signaling by removing K63-linked ubiquitin chains from the CARD domains of RIG-I and MDA5 [69], suggesting that different cellular means of RLR inactivation may be in play within the USP family.

The occurrence of many DUBs antagonizing K63-linked ubiquitination of RIG-I underscores the importance of this modification in the activation of the innate immune response. However, these DUBs might also have redundant functions and be active in different cell types or differ in their spatio-temporal activity, as some of them might play a role during uninfected conditions to prevent spontaneous activation of RLR signaling, while others could act specifically during the resolution phase of infection.

### 3.2. E3 ligases Destabilizing RLRs

#### 3.2.1. RNF122

Mass-spectrometry analysis of RIG-I interactors in virus-infected L929 cells identified the ER residing E3 ligase RNF122 as a negative regulator of RIG-I protein levels [76]. RNF122 expression is induced upon RNA virus infection and it was shown to interact with RIG-I both under basal conditions and upon infection with VSV [76]. This suggests a contribution to base-line RIG-I turnover, as well as increased removal upon immune stimulation. Consistent with this idea, BMDMs and MEFs from *Rnf122^−/−^* mice produced increased levels of IFNα and IFNβ, TNFα and IL6 in response to infection with VSV and SeV [76]. Accordingly, mice lacking *Rnf122* were less susceptible to infection with VSV, resulting from increased levels of type I IFNs in the serum. Mechanistically, RNF122 was shown to interact with RIG-I via its N-terminal transmembrane domain and to ubiquitinate RIG-I with K48-linked ubiquitin chains at K115 and K146 in its CARD domain [76].

#### 3.2.2. TRIM40

The E3 ligase TRIM40 was shown to attenuate both RIG-I and MDA5 dependent activation of type I IFN production [80]. In uninfected conditions, TRIM40 promoted proteasomal degradation of RIG-I/MDA5 by ubiquitination in their CARD domains with K48 and K27-linked ubiquitin chains, putatively as a protein quality control mechanism or a means to limit basal RLR levels. Upon virus infection, TRIM40 levels were downregulated, possibly to allow full activation of the immune response [80], supporting a model where it ‘keeps a foot on the break’, which is released upon infection. Accordingly, *Trim40*-deficient mouse peritoneal macrophages and MEFs had increased IFNβ, TNFα, and Interleukin 6 (IL6) expression in response to SeV and VSV infection, which was rescued by exogenous TRIM40 expression. Similarly, *Trim40*-deficient mice were resistant to VSV infection due to increased cytokine levels in the serum resulting in decreased viral replication [80].

#### 3.2.3. TRIM13

TRIM13 is upregulated in BMDMs upon stimulation with RLR agonists and was shown to specifically negatively regulate MDA5-induced IFN production [84]. *Trim13**^−/−^* mice had prolonged survival compared to their WT littermates due to increased type I IFNs production upon lethal infection with encephalomyocarditis virus (EMCV), which is sensed by MDA5. Similarly, *Trim13^−/−^* MEFs produced increased levels of IFNβ in response to MDA5 agonists. TRIM13 was shown to coimmunoprecipitate with MDA5, however no deeper mechanistic insight on how TRIM13 contributes to negative regulation of MDA5 dependent signaling is provided in this study [84].

However, it remains to be determined whether these effects are specific to MDA5, rather than a more general mechanism. This is fueled by the observation that *Trim13**^−/−^* MEFs also produced increased levels of IFNβ in response to RIG-I agonists, suggesting that it might promote RIG-I dependent signaling as well [84].

#### 3.2.4. RNF125

The RING E3 ligase RNF125 was identified as a negative regulator of several signaling molecules of the RIG-I signaling pathways, including RIG-I, MDA5, and MAVS [74]. RNF125 was shown to be induced by IFNs and to act together with the E2 enzyme Ubiquitin Conjugating Enzyme E2 D1 (UBE2D1/UBCH5), conjugating K48-linked ubiquitin chains to the N-terminal CARD domains of RIG-I [74]. Knock-down of RNF125 in Human Embryonic Kidney (HEK)-293FT cells decreased ubiquitination of RIG-I, whereas overexpression of RNF125 increased RIG-I ubiquitination, suppressing IRF3 activation and decreasing IFNβ promoter activity in a dose-dependent manner [74].

The broad range of possible ubiquitination targets described above, as well as other immune-related proteins [85,86,87], raises the question of how this is achieved. On the one hand, RNF125 could be targeting the activated complex of the RLRs and MAVS, resulting in their ubiquitination and degradation. Alternatively, RNF125 could be a more general immune-suppressive response gene acting on DUBs for these factors, or these effects could be indirect as RNF125 has other pleiotropic effects on TGFβ signaling [88].

Additionally, NLR Family Pyrin Domain Containing 12 (NLRP12) was shown to have important negative regulatory roles downstream of RLRs, TLRs, and NLRs. One study investigated the role of NLRP12 in RNA virus-induced RIG-I signaling [89]. *Nlrp12**^−/−^* BMDCs had increased TBK1 and IRF3 phosphorylation concomitant with increased production of type I IFNs upon infection with VSV or short dsRNA compared to WT cells. Accordingly, upon VSV infection, *Nlrp12* knock-out mice produced more IFNα and IFNβ resulting in lower VSV replication and decreased morbidity. The authors showed that NLRP12 interacted with TRIM25 and inhibited K63-linked ubiquitination of RIG-I and at the same time enhance its RNF125-dependent degradative ubiquitination, which reflected in lower RIG-I proteins levels in brain and spleen macrophages and DCs [89].

#### 3.2.5. CBL

The RING E3 ligase CBL was identified as an interactor of the lectin family member Sialic Acid Binding Ig Like Lectin (SIGLEC)-G, which is strongly induced by type I IFNs, and shown to be a negative regulator of RIG-I signaling by targeting the Src Homology 2 (SH2) domain–containing Phosphatase 2 (SHP2)/CBL complex to RIG-I [75]. This promotes RIG-I K48-linked ubiquitination at K813 and subsequent proteasomal degradation [75]. SIGLEC-G is further upregulated by type I IFNs, resulting in a negative feed-back loop of RIG-I signaling [75]. On the other side, CBL-dependent degradation of RIG-I is antagonized by F-Box and WD Repeat Domain Containing 7 (FBXW7), which upon SeV infection translocates from the nucleus to the cytoplasm and destabilizes SHP2 by K48-linked ubiquitination, thereby enabling downstream pro-inflammatory signaling [90]. The exact timing mechanisms driven these intersecting feedback loops remain to be identified. Moreover, the fact that SIGLEC-G is highly expressed in macrophages and DCs suggests that this suppressive mechanism may be specific to these cell types [91].

#### 3.2.6. STUB1/CHIP

The E3 ligase STUB1 (a.k.a. Carboxy Terminus Of HSP70-Interacting Protein (CHIP)) has important regulatory functions in protein quality control pathways by interacting with molecular chaperones and promoting the degradation of misfolded protein targets [92]. However, STUB1 was also identified to promote RIG-I K48-linked ubiquitination dependent on the Thrithorax family protein Lysine Methyltransferase 2E (KMT2E) [78]. The authors reported a novel role of KMT2E as a negative regulator of RLR-dependent type I IFN induction in vivo, as *Kmt2e*^−/−^ mice were significantly more resistant to VSV infection due to decreased viral replication resulting from increased IFNβ expression in liver and lung [78]. However, the negative regulatory role of KMT2E in proteasomal degradation of RIG-I mediated by STUB1 was modest, and alternative mechanisms of KMT2E-mediated immune regulation may be in place [78]. Although knock-down of *Stub1* in mouse peritoneal macrophages increased expression of IFNβ upon VSV and SeV infection [77], STUB1-dependent RIG-I ubiquitination may be a protein quality control mechanism rather than an immune-specific feed-back mechanism.

#### 3.2.7. LUBAC

The Linear Ubiquitin Assembly Complex (LUBAC), consisting of SHANK Associated Regulator of G-protein Signaling (RGS) homology (RH) Domain Interactor (SHARPIN) and the two E3 ligases Heme-Oxidized Iron Responsive Element Binding Protein 2 (IRP2) Ubiquitin Ligase 1 (HOIL1)-Interacting Protein (HOIP) and the long isoform of HOIL1 (HOIL-1L), has been proposed as another negative regulator of RIG-I-mediated induction of type I IFN signaling via two distinct mechanisms: Firstly, HOIL-1L competes with TRIM25 for the interaction with RIG-I, to suppress its K63-linked ubiquitination [12]. Secondly, HOIP and HOIL-1 were shown to induce TRIM25 degradation, attenuating RIG-I activation.

In vitro analysis showed that LUBAC induced the formation of linear ubiquitin chains on TRIM25 [12]. Moreover, ectopic LUBAC expression or SeV infection in cells induced linear and K48-linked ubiquitination of TRIM25, which was diminished upon knock-down of genes encoding LUBAC components [12]. Since LUBAC exclusively mediates linear ubiquitination of its targets [93], the K48-linked ubiquitination of TRIM25 is likely mediated by a yet unidentified E3 ligase recruited by LUBAC, possibly generating heterotypic ubiquitin chains [12]. The occurrence of K48-linked ubiquitin chains on TRIM25 is supported by another study, which identified USP15 as a positive regulator of RIG-I signaling, removing K48-linked ubiquitin chains from TRIM25 [94].

### 3.3. DUBs Inactivating MAVS

MAVS activation and aggregation, which is promoted by K63 ubiquitination catalyzed by TRIM31 [57], is counteracted by at least two DUBs of the OTU family. OTU Deubiquitinase 3 (OTUD3) was shown to remove K63-linked ubiquitin chains from MAVS, thereby suppressing MAVS aggregation and downstream activation of type I IFN signaling [95]. Accordingly, mice lacking *Otud3* had decreased morbidity after infection with VSV, resulting from increased production of cytokines and diminished viral replication [95]. OTUD3 activity was dependent on its acetylation at K129, which was shown to be removed by Sirtuin 1 (SIRT1) in response to RNA virus infection, allowing antiviral signaling [95]. However, the enzyme that mediates OTUD3 acetylation remains to be identified, as well as how OTUD3 activity is regulated in time.

### 3.4. E3 Ligases Destabilizing MAVS

MAVS acts as a critical adapter in RLR signaling, linking the cytoplasmic sensors RIG-I and MDA5 to their downstream signaling molecules. MAVS forms mitochondrial, prion-like aggregates as part of its activation mechanism [56]. This prompts the question how such a molecule is itself regulated through degradation and disassembly, and whether identified factors target the aggregate, or non-aggregate MAVS forms.

#### 3.4.1. MARCH5

MARCH5 is a mitochondrial residing E3 ligase, which has been reported to specifically target MAVS aggregates and thereby negatively regulate RLR-induced type I IFN production [96]. MARCH5 was shown to accumulate at the mitochondria and specifically interact with MAVS upon RIG-I stimulation, thereby decreasing the amount of MAVS aggregates [96]. Accordingly, knockout of *March5* increased the number of MAVS aggregates upon stimulation with RIG-I agonists, a phenotype which could be reversed upon exogenous expression of WT, but not the RING mutant MARCH5^H43W^.

Moreover, *March5**^−/−^* Raw264.7 cells and *March^+/−^* BMDMs produced increased type I IFNs in response to infection with different RNA viruses, concomitant with lower viral replication [96]. This was reflected in vivo, as *March5^+/−^* mice were less susceptible to infection with VSV compared to their WT littermates [96]. Mechanistically, MARCH5 was shown to induce K48-linked ubiquitination of MAVS at K7 and K500 thereby promoting its proteasomal degradation [96], although how the specific interaction with MAVS upon stimulation is achieved remains to be determined. It could be that MARCH5 has a broader ubiquitination effects in the activated RLR-MAVS complex, as a recent study showed that MARCH5 also conjugated K48-linked polyubiquitin chains to RIG-I at K193 and K203 in its CARD domain [79].

#### 3.4.2. Itchy E3 Ubiquitin Protein Ligase (ITCH)

The HECT domain E3 ligaseITCH, which is recruited to the mitochondria in response to viral infection, was identified as a negative regulator of RLR-signaling, inducing MAVS degradation by ubiquitination with K48-linked ubiquitin chains at K371 and K420 [97]. Knock-down of ITCH increased the resistance of cells to Newcastle disease virus (NDV) infection due to increased RLR-mediated signaling. The activity of ITCH was shown to be dependent on Poly(RC) Binding Protein 2 (PCPB2), which functions as an adapter between ITCH and MAVS [97]. Later studies identified Poly(RC) Binding Protein 1 (PCBP1) to mediate MAVS degradation under basal conditions, thus possibly playing a role in either protein quality control, or maintenance of steady-state MAVS levels [98].

NLR Family Member X1 (NLRX1) was shown to inhibit the association between RIG-I and MAVS and TRAF6-dependent IKK activation in response to viral or bacterial infection [99]. *Nlrx1^−/−^* mice produced increased levels of IFNβ and antiviral response genes, concomitant with decreased viral titers in lung homogenates in response to influenza virus infection. Another study suggested the involvement of NLRX1 in the degradation of MAVS depending on the adapter protein PCBP2, possibly via recruitment of the E3 ligase ITCH [97,100]. The negative regulatory role of NLRX1 was questioned by another study that failed to identify changes in IFN and NFκB signaling in *Nlrx1^−/−^* mice upon intravenous poly(I:C) injection [101].

Lastly, Tax1 Binding Protein 1 (TAX1BP1) was reported as an adapter protein for ITCH, targeting ITCH to MAVS and promoting its degradation by ubiquitination at K10 both under basal and infected conditions [102], again suggesting that this degradation may not specifically resolve the induced form of MAVS.

#### 3.4.3. TRIM25

TRIM25, an activator of RLR-induced cytokine N production by promoting RIG-I K63 ubiquitination [50], was shown to also negatively regulate RLR signaling in a RIG-I-independent way, as stimulation of *Trim25**^−/−^* MEFs with the MDA5 agonist poly(I:C) significantly inhibited IFNβ production [103]. The authors proposed a model in which TRIM25 catalyzes first the activation of RIG-I by ubiquitination with K63-linked ubiquitin chains, followed by its association with MAVS, which acts as a platform to recruit downstream signaling complexes.

Subsequent K48-ubiquitination of MAVS at K7 and K1094 by TRIM25 allows the release of the associated signaling complex to the cytoplasm, enabling downstream signaling [103]. It will be interesting to understand, how these two spatially separated mechanisms are temporally controlled.

Moreover, it remains to be determined, how TRIM25 is directed to synthesize two different poly-ubiquitin linkages. Given that TRIM25 is a RING E3 ligase, it is possible that recruitment of different E2 enzymes account for the differential poly-ubiquitin linkages. As such, differential usage of E2 enzymes by an activating E3 ligase, could be an effective cellular strategy to time switching from immune-promoting to immune-resolving responses. Alternatively, mono-ubiquitin deposited on MAVS by TRIM25, could be extended with K48 chains by a yet unknown secondary E3 ligase.

#### 3.4.4. TRIM29

Epithelial cells are constantly exposed to external micro-organisms. At these sites, specific cellular mechanisms are in place to ensure that these potential invaders are kept in check, as well as mechanisms to prevent excessive activation by this incessant exposure.

TRIM29 is an E3 ligase constitutively expressed in the airway epithelium [104] and was shown to be induced by dsDNA in myeloid DCs [105]. Knockout of *Trim29* increased survival rates of mice after virus infection, due to increased production of type I IFNs, which restricted viral replication [105]. Mechanistically, TRIM29 was shown to interacted with MAVS upon viral infection and to mediate its ubiquitination with at the K11-linked ubiquitin chains at K371, K420, and K500 [105]. This suggests that TRIM29 may play a specific suppressive role in activated epithelial cells, although how specific interaction with activated MAVS is achieved remains to be determined.

#### 3.4.5. Sma and Mothers Against Decapentaplegic Homolog (SMAD) Specific E3 Ubiquitin Protein Ligase (SMURF)1/2

Another study identified the HECT domain E3 ligase SMURF2 to ubiquitinate MAVS with K48-linked ubiquitin chains and promote its degradation. Overexpression of WT, but not E3-ligase deficient SMURF2 in HEK293T cells inhibited SeV-induced IFNβ production and e Interferon-Stimulated Response Element (ISRE) activation, whereas *Smurf2* knock-out in MEFs enhanced SeV-induced IFNβ and ISG15 expression concomitant with increased MAVS protein stability [106]. Similarly, SMURF1, depending on the Neural Precursor Cell Expressed, Developmentally Down-Regulated 4 (NEDD4) family-interacting protein 1 (NDFIP1), was reported to mediate MAVS ubiquitination [107].

The DUB OTUD1 was further identified as a negative feed-back regulator of RLR signaling, being strongly induced upon RNA virus infection and stabilizing SMURF1 by removing its ubiquitination. Mice lacking OTUD1 had increased resistance to RNA virus infection due to elevated IFNβ production and decreased viral replication. Upon RNA-virus infection, OTUD1 deubiquitinated SMURF1, which promoted association with the MAVS-TRAF3-TRAF6 signalosome, and enabled the degradation of the single components [108].

### 3.5. DUBs Inactivating TBK1

The serine and threonine kinase TBK1 is a node point in the type I IFN induction pathways, where signals from various immune adapter proteins including MAVS, the Toll/Interleukin-1 Receptor (TIR)-domain-containing Adapter-inducing Interferon-β (TRIF; a.k.a. Toll Like Receptor Adaptor Molecule 1 (TICAM)) and Stimulator of Interferon Genes (STING) converge, resulting in the activation of the transcription factors IRF3 and IRF7 [109]. In this section we discuss mechanisms of TBK1 suppression; the involvement of ubiquitin in the regulation of its downstream targets IRF3 and IRF7 is discussed in the section ‘Negative regulation of IFN signaling and induced response proteins’.

TBK1 activity is controlled by post-translational modifications (PTM), amongst them K63-linked ubiquitination [63,64,65]. Several DUBs have been reported to antagonize this modification. CYLD was shown to remove K63-linked ubiquitin chains from several signaling molecules within the RIG-I signaling pathway including TBK1, suggesting that it might be a general negative regulator of type I IFN induction [67]. Knockdown of *Cyld* increased SeV-induced IRF3 and IκBα phosphorylation and IFNβ mRNA expression [67]. Importantly, CYLD levels were downregulated upon SeV infection in the presence of TNF, likely to allow full activation of IFN signaling [67].

In addition, the DUB A20 was identified as a negative regulator of TBK1 K63-linked ubiquitination [110]. The recruitment of A20 to the TBK1/IKKε complex was shown to be dependent on the adapter protein TAX1BP1, which had been shown previously to be important for the regulation of the activity of A20 in the context of NFκB signaling [111,112]. However, WT and DUB domain (C103A) mutant A20 equally inhibited TBK1/IKKε ubiquitination, indicating that A20 acted in a deubiquitinase-independent manner by disrupting the interaction between TRAF3 and the TBK1/IKKε complex [110].

### 3.6. E3 Ligases Destabilizing TBK1

Several E3 ligases have been reported to destabilize TBK1 by promoting its proteasomal degradation.

#### 3.6.1. DTX4

Firstly, Deltex E3 Ubiquitin Ligase 4 (DTX4) was reported to mediate TBK1 ubiquitination with K48-linked ubiquitin chains at K670 [113]. DTX4 recruitment to TBK1 was shown to be dependent on the NOD protein family member NLRP4, which interacted specifically with activated (phosphorylated) TBK1 upon viral infection [113]. Consistently, knock-out of either *Dtx4* or *Nlrp4* decreased K48-ubiquitination of virus-activated TBK1 [113]. A subsequent study identified the kinase Dual Specificity Tyrosine Phosphorylation Regulated Kinase 2 (DYRK2) to promote DTX4-dependent degradation of TBK1 by phosphorylating S527 of TBK1, which is essential for the recruitment of DTX4 by NLRP4 [114]. TBK1 degradation was promoted by viral infection, and inhibition of protein synthesis with the drug cycloheximide decreased the protein half-life under infected conditions, suggesting DTX4-dependent degradation of TBK1 to be a negative feed-back mechanism to negatively regulate type I IFN production [113].

Recently, also TRAF3 Interacting Protein 3 (TRAF3IP3) was identified to recruit DTX4 to TBK1 and to promote TBK1 ubiquitination at K372 [115]. *Traf3ip3*-deficient BMDMs had decreased K48-ubiquitination of endogenous TBK1 in response to VSV infection, were more resistant to VSV infection, and had decreased viral titers due to increased IFNβ production [115]. In addition to promoting TBK1 degradation, TRAF3IP3 was shown to compete with MAVS for TRAF3 and TBK1 binding, thereby inhibiting downstream signaling [115].

#### 3.6.2. TRAF Interacting Protein (TRAIP)

The RING-type E3 ligase TRAIP was shown to promote TBK1 degradation by ubiquitination with K48-linked ubiquitin chains [116]. TRAIP is strongly induced in macrophages upon stimulation with TLR agonists, and was shown to inhibit Lipopolysaccharide (LPS)- and poly(I:C) induced IRF3 activation [116]. Accordingly, knock-down of TRAIP in primary peritoneal macrophages increased IRF3 activation upon infection with SeV, consistent with increased IFNβ production resulting from increased TBK1 activity [116]. Degradation of TBK1 by DTX4 and TRIP is promoted by the DUB USP38 [117]. Following virus infection, USP38 is recruited to the activated TBK1 by NLRP4 and removes K33-linked ubiquitin from K670, which enables K48-ubiquitination by DTX4 and TRIF [117].

#### 3.6.3. TRIM27

In addition, TRIM27 was shown to negatively regulate virus-induced type I IFN production by promoting TBK1 proteasomal degradation through ubiquitination at K251 and K372 [118]. Similar to CBL, which negatively regulates type I IFN production by targeting RIG-I dependent on the lectin family member SIGLEC-G [75], TRIM27 activity was shown to be dependent on the lectin family member SIGLEC1, which is strongly induced by Janus Kinase (JAK)/STAT1 signaling and associates with DNA polymerase III gamma/tau (DNAX)-Activation Protein 12 (DAP12) and SHP2 to recruit TRIM27 and promote TBK1 ubiquitination, resulting in a negative feedback loop to downregulate type I IFN production [118].

### 3.7. DUBs and E3 Ligases Inhibiting TRAF3 Function in Type I IFN Production

#### 3.7.1. OTUD5/Deubiquitinating Enzyme A (DUBA)

OTUD5/DUBA was identified as a negative regulator of the *Ifna4* promoter in an siRNA screen for OTU DUB family members involved in the negative regulation of type I IFN and cytokine production [119]. Silencing *Otud5* in HEK293 cells increased type I IFN and Regulated Upon Activation, Normally T-Expressed, And Presumably Secreted (RANTES) expression in response to infection with SeV, and poly(I:C) stimulation. OTUD5 was shown to interact with TRAF3 upon poly(I:C) stimulation and to remove K63-linked ubiquitin chains, promoting its dissociation from the TBK1 signaling complex and inhibiting type I IFN production. As OTUD5 is induced by LPS stimulation, it might act as a negative feedback regulator of IFN signaling by targeting TRAF3 [119].

#### 3.7.2. MYB-like, Swi3p, Rsc8p and Moira (SWIRM) and MPN Domains 1 (MYSM1)

The DUB MYSM1 has been identified as a histone H2A deubiquitinase [120], with important regulatory functions in hematopoiesis and immune signaling [121]. In this context, its importance in various PRR signaling cascades activated by RLRs, TLRs, and cytoplasmic DNA sensors was demonstrated in vivo, as *Mysm1^−/−^* mice were more susceptible to viral infection compared to WT mice, showing a hyper-inflammatory immune response enabling faster viral clearance but having a higher morbidity [122]. Mechanistically, MYSM1 translocated to the cytoplasm upon infection to remove K63-linked ubiquitin chains from TRAF3 and TRAF6, terminating type I IFN induction, before being rapidly degraded itself [122]. It will be interesting to determine, how MYSM1 localization and activity are regulated, given that nuclear MYSM1 was constitutively present, while cytoplasmic MYSM1 was rapidly turned-over [122].

#### 3.7.3. RNF216

RNF216 (a.k.a. Two RING fingers and double RING finger linked (DRIL) (TRIAD) Domain-Containing Protein 3 (TRIAD3A)), previously implied in targeting several TIR-domain containing proteins including TLR4, TLR9, and TRIF for proteasomal degradation [123,124], was also proposed to promote TRAF3 degradation by K48-linked ubiquitination [125]. RNF216 was induced by RNA virus infection and its expression resulted in degradation of endogenous TRAF3. Conversely, knock-down of *Rnf216* increased TRAF3 expression under basal conditions, while increasing type I type I IFN and ISG levels during SeV infection, suggesting that RNF216 may contribute both to TRAF3 steady-state protein turnover, but also during infection as part of a negative feedback mechanism [125].

## 4. Negative Regulation of TLR Signaling

TLRs are a family of transmembrane proteins capable of recognizing a wide range of PAMPs such as lipids, lipoproteins, lipopolysaccharides, and nucleic acids. Humans encode 10 different TLRs, of which TLR1, 2, 4, 5, 6, and 10 are expressed at the plasma membrane and TLR 3, 7, 8, and 9 in endosomes (Figure 2) [126].

Upon activation, TLRs signal via distinct TIR-domain containing adapter proteins such as MYD88, TRIF, Translocation Associated Membrane Protein (TRAM), and TIR Domain Containing Adaptor Protein (TIRAP). All TLRs, except for TLR3, signal via the adapter MYD88, which recruits Interleukin 1 Receptor Associated Kinase (IRAK) kinases to activate the ubiquitin ligase TRAF6 (Figure 2) [126]. TRAF6 undergoes autoubiquitination with K63-linked ubiquitin chains, which are required to recruit the kinase complexes TAK1/TAB1/2/3, and the heterotrimeric kinase complex consisting of IKKα/β/NEMO. K63-linked ubiquitination by TRAF6 activates TAK1, which phosphorylates IKKα/β, resulting in NFκB-dependent transcription of proinflammatory genes [127,128]. Moreover, conjugation of linear ubiquitin chains by LUBAC to NEMO is required for TAK1-dependent activation of the IKK complex [129,130].

TLR3 and TLR4 engage TRIF to recruit TRAF3, which undergoes K63-linked autoubiquitination similarly to TRAF6, required for the recruitment of NEMO/TBK1/IKKε [131] and IRF3/IRF7-dependent induction of type I IFNs. In addition, TRIF signals via TRAF6, which recruits the kinase RIP1 to promote TAK1/TAB1/2/3-dependent activation of NFκB signaling [127,128].

Most pro-inflammatory pathways converge onto a common kinase layer (TAB/TAK, and classic IKK complex), which drives NFκB activation. This layer is substantially controlled by ubiquitination, which is discussed in more detail in the TNF signaling section. Upstream of this common signaling hub, TLRs themselves, and also their TIR domain containing adaptors are substantially negatively regulated in different ubiquitin-dependent manners. Several activation-induced feedback mechanisms have been reported, contributing to curtailing TLR-specific responses.

Many of these cellular mechanisms interfere with functional assembly of TIR-domain-mediated signaling complexes of the TLRs and their adaptors. In particular, MYD88 is essential of signaling of all TLRs except TLR3, and was recently shown to form large helical filaments which function as signaling scaffolds, and subsequently recruit IRAK members for signal propagation [132]. Biological output of this Myddosome is restricted by its disassembly [132,133], and restriction of TIR protein concentrations through degradation. A full summary of reported negative regulators can be found in Table 1; below we discuss several specific regulators of the TLR pathway acting on the activated pool of TLRs or their adaptors, or putatively mediate their turnover as part of protein quality control (QC).

### 4.1. DUBs Regulating TLR and TLR-Adaptor Activity

#### DUBs Deactivating MYD88

MyD88 activity has been proposed to be antagonized by two DUBs, which remove K63-linked ubiquitin chains from MYD88: OTUD4 and CYLD. OTUD4 has been associated with cerebellar ataxias [134], although the mechanism behind this remains currently undefined. Although at this point no clear immune-disease association has been established for OTUD4, there is increasing evidence from knock-out studies for a role in suppressing TLR-induced inflammation [135].

OTUD4 was identified by mass spectrometry analysis to interact with MYD88 and to target K63-linked ubiquitin chains of MYD88, depending on phosphorylation at S202 and S204 and a previously unknown ubiquitin-interacting motif [135]. Knock-out of *Otud4* in MEFs increased NFκB signaling upon stimulation with IL1β. Moreover, macrophages from *Otud4^−/−^* mice produced increased TNFα and IL6 following LPS stimulation [135]. Importantly, knock-out of *Otud4* in MEFs increased MYD88 K63-linked ubiquitination, but also IRAK1 and TRAF6, indicating that OTUD4 might be a more general negative regulator of NFKB signaling [135].

In addition to OTUD4, the DUB CYLD has been long recognized as an essential DUB for curtailing inflammatory responses, and controlling cell death [136]. In line with this notion, *Cyld* mutations have been linked to various immune-related diseases [136]. While it has become increasingly clear that CYLD is in this context essential for controlling TNF signaling, it likely plays important roles in TLR-mediated responses during bacterial infection as well, specifically removing activating K63 poly-ubiquitin chains on MYD88 [137].

In this context, CYLD was shown to target K63-linked ubiquitin chains on MYD88, which are induced by infection with *Haemophilus influenzae* (NTHi) in vitro and in vivo [137]. Overexpression of CYLD in cells decreased K63-ubiquitination levels, whereas its knock-down increased K63-linked ubiquitination of MYD88 upon infection with NTHi [137]. Importantly, the same was detected in lung tissue lysates of *Cyld*-deficient mice [137], underpinning the importance of CYLD as an inhibitor of the inflammatory response. Moreover, infection with NTHi resulted in higher production of IL1β, IL6, and TNFα in *Cyld^−/−^* mice compared to WT mice. Mechanistically, CYLD was shown to remove K63-linked ubiquitin from K231 in the TIR domain of MYD88 [137].

### 4.2. E3 Ligases Regulating TLR and TLR-Adaptor Turn-Over

#### 4.2.1. RNF216

The ubiquitin ligase RNF216 was identified in a yeast two-hybrid screen to interact with the cytoplasmic TIR-domain-containing tail of several TLRs [123]. Depletion of Rnf216 increased TLR expression and signaling in response to the respective ligands of TLR4 and TLR9 in Human Umbilical Vein Endothelial Cells (HUVEC) and HEK-293T cells, resulting from diminished ubiquitination and proteasomal degradation [123]. Importantly, RNF216-dependent degradation of these TLRs was activation-dependent [123], suggesting a specific negative feedback loop for curtailing the inflammatory response. However, how this degradation is regulated to be activation-dependent, and how this is regulated in cellular space remains to be investigated.

Importantly, TLR2 was not ubiquitinated and degraded by RNF216 [123], indicating specificity in TIR domain recognition. Further studies showed that overexpression of RNF216 also negatively regulates the abundance of the TIR-domain containing adapter proteins TRIF and TIRAP, and the protein RIP1, which is suggested to have a TIR-homologous domain [124].

Interestingly, chemical Heat Shock Protein 90 (HSP90) inhibition further destabilized RIP1 in a RNF216-dependent manner [124], which raises the question whether some of the degradation of these adapters is part of protein QC, or whether HSPs act as molecular switches by binding these TIR-containing proteins under specific conditions. One should keep in mind that, HSP90-paralogs HSP90B1 (a.k.a. GP96) and CNPY3 (a.k.a. PRAT4A) are essential for TLR movement through the ER to their final membrane locations [138], and their potential inhibition by chemical HSP90 inhibitors may thus indirectly influence activation, and subsequently degradation of downstream adaptors.

#### 4.2.2. RNF170

Recently, the ubiquitin ligase RNF170 was identified by mass spectrometry analysis of TLR3-binding proteins in mouse peritoneal macrophages as a negative regulator of TLR expression [139]. Unlike RNF216 described above, RNF170 colocalized with TLR3 at the ER under non-stimulated conditions, and both proteins translocated to the early endosome upon poly(I:C) stimulation [139]. Mechanistically, RNF170 was shown to promote TLR3 degradation by K48-linked ubiquitination of K766 in its TIR domain [139].

Although Rnf170^−/−^ mice and their derived myeloid cells produced more cytokines upon TLR3 stimulation, and were consequently more resistant to viral infection [139], it remains unknown how RNF170 is itself activated. The provided data, and relatively modest effect of Rnf170 knock-out on viral levels and survival [139], may suggest that this E3 controls basal TLR levels, although this remains to be studied in further detail.

#### 4.2.3. Casitas B-Lineage Lymphoma Proto-Oncogene B (CBLB)

The E3 ligase CBLB was identified to negatively regulate TLR4 signaling [140]. *Cblb*-deficiency in mice resulted in acute lung inflammation and increased mortality after induction of poly-microbial sepsis, concomitant with increased cytokine and chemokine production [140]. CBLB was shown to be important for the downregulation of TLR4 after initiation of TLR4 stimulation, and for the association with the adapter MYD88 [140]. Mechanistically, CBLB may negatively regulate TLR signaling by ubiquitinating both adapter proteins MYD88 and TRIF with K48-linked ubiquitin chains, promoting their proteasomal degradation [141].

The interaction of CBLB with MyD88 and TRIF was shown to be dependent on integrin Subunit alpha M (ITGAM) (a.k.a. CD11B), which recruits the kinases Spleen Associated Tyrosine Kinase (SYK) and SRC to phosphorylate MYD88 and TRIF, creating docking sites for CBLB [141]. In mice, *Itgam*-deficiency enhanced TLR-mediated inflammatory responses, rendering mice more susceptible to pathogen-caused sepsis [141]. Together, these findings suggest an activation-dependent mechanism that curtails TLR adaptor levels through generation of a phospho-degron.

#### 4.2.4. SCF^SPOP^

The mechanisms for TLR suppression described above play predominant roles in myeloid cells to dampen their activation. However, it is equally important to ensure that cells in which immune responses should in principle be limited, have mechanisms in place to restrict expression of TLRs and their adaptors. In this context, the Cullin E3 ubiquitin ligase adaptor Speckle Type BR-C, ttk and bab (BTB)/Pox virus and Zinc finger (POZ) Protein (SPOP) was identified to restrict inflammatory responses in hematopoietic stem cells (HSC) [142]. SPOP recognizes S/T rich degron motifs [143], which can be mediated or inhibited by phosphorylation of these residues depending on the protein context [143,144].

Conditional knock-out of *Spop* in HSCs resulted in acute and lethal neutrophilia in mice upon sub-lethal LPS or poly(I:C) challenge [142]. Moreover, *Spop*-deficient HSCs were unable to resolve inflammatory responses upon poly(I:C) stimulation [142]. SPOP was shown to interact with a putative SPOP-degron of MYD88, and destabilize MyD88 by ubiquitination [142]. Together, these findings support the notion that SPOP limits the basal levels of several TLRs, thereby preventing premature activation in the HSC state, and their differentiated myeloid forms.

Consistent with this idea, conditional knock-out of *Spop* rendered mice more susceptible to Salmonella infection, and increased proinflammatory cytokine levels in the serum and in BMDMs, concomitant with MYD88 accumulation [145]. The same authors showed that chicken SPOP limits cytokine production by interacting with MYD88, thereby promoting its K48-linked ubiquitination at K118, K124, and K143 [145]. In line with this notion, another study found that human SPOP negatively regulates NFκB signaling through ubiquitination of MYD88, which is abolished by frequent mutations of both proteins occurring in several lymphoid malignancies [146].

Data from other studies may suggest that SPOP-dependent MYD88 degradation could be cell type dependent, or be influenced by the experimental conditions. For example, contrary to the described mechanism above, *Spop* depletion did not destabilize MYD88 in several other cell lines [146], whereas co-expression of SPOP and MYD88 robustly ubiquitinated MYD88 with mixed-ubiquitin chains at K203, K263, and K269 [146]. The authors proposed that negative regulation of NFκB activity by SPOP derives mainly from inhibition of MYD88-IRAK1 association, rather than from MYD88 destabilization [146]. What underlies this discrepancy will need to be determined in further studies.

#### 4.2.5. WW Domain Containing E3 Ubiquitin Protein Ligase (WWP)1/2

WWP2 is a NEDD4-like E3 ligase associated with regulating Phosphatase and Tensin Homolog (PTEN) ubiquitination and degradation, the dysregulation of which contribution to tumorous growth in Cowden syndrome [147]. However, a *Wwp2* knock-out mouse model also displayed increased cytokine production, which suggested additional roles in negative regulation of the innate immune response [148].

WWP2 was identified by shotgun mass spectrometry analysis of TRIF-interacting proteins [148]. In reporter gene assays WWP2 specifically inhibited TRIF-dependent, but not MYD88-dependent signaling, by targeting TRIF with K48-linked ubiquitin chains [148]. Consistent with these cell-based findings, macrophages from *Wwp2^−/−^* mice produced increased levels of IFNβ, C-C Motif Chemokine Ligand 5 (CCL5), TNFα, and IL6 upon stimulation with the TLR3-ligand poly(I:C) [148]. Accordingly, mice deficient for *Wwp2* were more susceptible to poly(I:C)-induced death compared to their WT littermates, concomitant with enhanced cytokine levels in their sera [148].

The related family member WWP1 has been implicated in curtailing TLR signaling as well [149], albeit with an effect on TRIF-independent signaling through TLR4. *Wwp1* knock-down reduced TRAF6 K48 poly-ubiquitination [149], suggesting the possibility of a TLR-common down-stream regulatory node, although the biological contribution of WWP1 at an organismal level will need further investigation in mouse models.

## 5. Negative Regulation of Cytosolic DNA Sensor Pathways

Cytosolic dsDNA is recognized by the sensor cyclic-GMP-AMP synthase (cGAS), which catalyzes the synthesis of cyclic GMP-AMP (cGAMP), a second messenger that activates the endoplasmic reticulum (ER)-localized adapter protein STING (also known as Mitochondrial Mediator of IRF3 Activation (MITA), Endoplasmic Reticulum IFN Stimulator (ERIS), Methionine-Proline-Tyrosine-Serine Plasma Membrane Tetraspanner (MPYS)) (Figure 1) [150]. Additional cytoplasmic DNA sensors such as Z-DNA Binding Protein 1 (ZBP1), Interferon Gamma Inducible Protein 16 (IFI16) and DEAD-Box Helicase 41 (DDX41) have also been described to recognize cytoplasmic DNA and activate STING, however it is not clear whether their function is cell type specific; to date predominantly cGAS has met with universal acceptance [151,152,153].

In addition to cGAMP-generated cyclic di-nucleotides, the STING signaling axis is also activated by other stimuli such as cyclic di-nucleotides produced by bacteria, mitochondrial DNA, cytosolic chromatin, and micronuclei [154]. Activated STING translocates from the ER to the Golgi, where it recruits TBK1 and multimerizes, forming the STING-signalosome [155,156]. STING activates TBK1, which phosphorylates itself, STING and IRF3, leading to the production of type I IFNs [157]. Moreover, STING utilizes an unconventional, TRAF-independent signaling route to activate NFκB and Mitogen-Activated Protein Kinase (MAPK) [158]. cGAS and STING were shown to be heavily regulated by post-translational modifications, including phosphorylation, sumoylation, and ubiquitination, which modify their activity, trafficking, and stability [158].

### DUBs and E3 Ligases Controlling cGAS/STING Activity and Turn-Over

At the receptor level, the E3 ligase TRIM21, which is strongly induced by type I IFNs, was shown to target the DNA sensor DDX41 at K9 and K115 with K48-linked ubiquitin chains, thereby negatively regulating dsDNA-triggered signaling in myeloid DCs and monocytes [159]. *Trim21*-deficient mice produced increased type I IFNs and IL6 in response to HSV-1, consistent with significantly decreased viral titers [159]. In addition, TRIM21 was reported to negatively regulate IFI16-dependent type I IFN induction [159]. The authors proposed a negative feed-back model in which STING recruits TRIM21, which targets IFI16 at K3, K4, K6 with K48-linked ubiquitin chains to prevent overproduction of type I IFN [160].

Not much is currently known about the role of ubiquitination in the negative regulation of cGAS. It was proposed that ubiquitination of cGAS at K414 with K48-linked ubiquitin chains promotes recognition by p62 and autophagic degradation, which is antagonized by TRIM14, an ISG upregulated upon viral infection, which recruits the DUB USP14 to remove K48-linked ubiquitin and prevent cGAS degradation [161]. However, the E3 ligase targeting cGAS with K48-linked poly-ubiquitin remains to be determined.

At the STING adaptor level, especially ubiquitination at K150 has been reported to be crucial for STING activation [162,163,164,165]. Multiple E3 ligases, including RNF185, Mitochondrial E3 Ubiquitin Protein Ligase 1 (MUL1), TRIM32, and TRIM56 have been implicated in this event [162,163,164,165], indicating that they may have redundant roles, or act in a stimulus- or cell type-specific context. In support of a key role of K150 in STING activity, several antagonistic mechanisms have been reported. Firstly, CYLD was shown to remove K48-linked ubiquitin from K150 of STING, stabilizing its expression and promoting induction of IRF3-responsive genes [166].

Moreover, the E3 ligase RNF5 was identified in a Y2H screen as an interactor of STING [167], targeting K150 of STING with K48-linked ubiquitin chains, and thereby inducing its proteasomal degradation. Interestingly, this study found localization of part of the STING pool at the mitochondria, which is where sub-cellular fractionation experiments suggested RNF5 drives STING ubiquitination and degradation [167]. It should be noted that RNF5-STING interaction could be driven independently of STING activation by SeV infection, which may suggest that IRF3- or type I IFN-dependent responses are crucial for generating an activation-induced feedback loop [167]. However, RNF5 also inhibited SeV-induced, STING-independent, type I IFN induction [167], prompting the necessity for further studies to address RNF5 pathway specificity.

RNF5-dependent STING degradation may be countered by its family member RNF26 [168]. RNF26 was shown to conjugate K11-linked ubiquitin to K150 of STING, likely interfering with K48-linked ubiquitination mediated by RNF5 and protecting it from degradation, thereby promoting type I IFN production [168]. However, in a later phase of viral infection RNF26 might also have an inhibitory role, promoting autophagic degradation of IRF3 [168].

In addition, the E3 ligase TRIM29 has been shown to promote the degradation of STING in the airway epithelium [104]. TRIM29 is highly expressed in human airway epithelial cells and further induced by EBV infection in human DCs [104]. TRIM29 was shown to induce K48-linked ubiquitination of STING at K370, promoting it proteasomal degradation. *Trim29*-deficiency in BMDMs and BMDCs resulted in enhanced production of type I IFN and IL6 in response to stimulation with dsDNA viruses or STING-agonists [104].

Moreover, two independent studies reported that *Trim29^−/−^* mice are resistant to DNA infection due to enhanced production of type I IFNs, and have decreased virus titers in the lung [104,169]. Another study extended these results showing that TRIM29 is highly induced in response to cytoplasmic DNA in several cell types [169]. The authors proposed that upon virus-infection TRIM29 interacts with STING at the perinuclear region, promoting its degradation by K48-ubiquitination at K288 and K337.

Lastly, mouse-specific TRIM30α has been reported to promote STING degradation in DCs, by targeting K275 with K48-linked ubiquitin [170]. *Trim30^−/−^* DCs produced elevated type I IFNs and IL6 in response to DNA viruses [170]. In line with this notion, *Trim30*-deficiency increased survival to infection with HSV-1 in mice [170]. *Trim30* is mouse-specific, which raises the question whether the closest human ortholog—*Trim5*—has similar functions in humans.

Given that STING activity is regulated by ubiquitination, it is feasible that this modification is antagonized by deubiquitinating enzymes. To our knowledge, to date only USP13 has been identified to target STING. USP13 has been shown to remove K27-linked ubiquitin chains from unidentified lysine residues in STING, thereby preventing the recruitment of TBK1 [171]. Indeed *Usp13^−/−^* mice were more resistant to infection to HSV-1 [171]. Similarly, BMDCs and MEFs derived from *Usp13^−/−^* mice displayed enhanced expression of type I IFNs and proinflammatory cytokines in response to HSV-1 infection [171].

## 6. Negative Regulation of NLR Signaling

Nucleotide-binding oligomerization domain (NOD)-like receptors are cytoplasmic sensors that recognize a wide range of microbial- and danger-associated molecular patterns such as bacterial cell wall components, toxins, or other cytosolic danger signals [172]. NOD-like protein share a tripartite structure: (**i**) an N-terminal effector domain, for instance a CARD domain in NOD1 and NOD2, which is required for the signaling to downstream effector molecules; (**ii**) a central nucleotide-binding domain (NBD, NOD, or NLP family apoptosis inhibitor protein (NAIP), C2TA or MHC class II transcription activator (CIITA), incompatibility locus protein Podospora anserina (HET-E), and TP1 or telomerase-associated protein (TEP1) (NACHT) domain); (**iii**) and a series of C-terminal leucine-rich repeats (LRR) involved in ligand sensing [173,174]. Several members of the NLRP subfamily such as NLRP1 and NLRP3 are involved in inflammasome formation and activation of inflammatory caspases, promoting the maturation of pro-inflammatory cytokines IL1β and IL-18 (reviewed in [173]).

In recent years, it has become clear that NOD1 and NOD2 are involved in the initiation of innate immune responses in response to pathogens [172]. NOD1 and NOD2 are two well-characterized members of the Nucleotide-Binding Oligomerization Domain, Leucine Rich Repeat And CARD Domain Containing (NLRC) subfamily that trigger the activation of NFκB and production of proinflammatory cytokines [175,176]. Mutations in NOD1/2 are associated with inflammatory disorders, such as inflammatory bowel disease or Crohn’s disease [177,178,179].

Sensing of cytosolic peptidoglycan by NOD1 or NOD2 induces the formation of the NOD-signaling complex (NOD-SC): NOD1 and NOD2 oligomerize via their NOD-domain and recruit the kinase Receptor Interacting Serine/Threonine Kinase 2 (RIP2) through CARD–CARD interactions (Figure 2) [180,181]. Hereupon, several E3 ligases, amongst them X-Linked Inhibitor of Apoptosis (XIAP) and cellular Inhibitor of Apoptosis Protein 1/2 (cIAP1/2), are recruited to RIP2 and ubiquitinate RIP2 with K63-linked ubiquitin chains, promoting the recruitment of LUBAC [182,183,184]. Synthesis of M1 and K63-linked poly-ubiquitin on RIP2 further induces the recruitment of the TAK1/TAB2/3 and IKKα/IKKβ/NEMO complexes [185]. NEMO is further ubiquitinated by HOIP with linear ubiquitin chains and TAK1 with K63-linked ubiquitin chains, eventually mediating activation of NFκB [183,184,186].

### 6.1. DUBs Curtailing NOD1/2 Signaling

Several DUBs have been reported to negatively regulate NOD signaling, counteracting the ubiquitination of components of the NOD-SC. A20 (encoded by the Tnfaip3 gene) negatively regulates NOD-SC mediated NF-kB activation by deubiquitinating RIP2 [187]. Tnfaip3-deficiency increased production of proinflammatory cytokines in response to stimulation with the bacterial cell-wall peptidoglycan Muramyl Dipeptide (MDP) both in primary macrophages and in mice, resulting from increased RIP2 ubiquitination [187].

Similarly, the DUB Ovarian Tumor Deubiquitinase With Linear Linkage Specificity (OTULIN) which hydrolyses linear ubiquitin chains synthesized on RIP1 by LUBAC in TNF-signaling pathways [188], was identified to restrict linear ubiquitination of RIP2, attenuating signaling downstream of NOD2 [185]. Accordingly, overexpression of WT, but not catalytically-inactive or M1-ubiquitin-binding-deficient OTULIN inhibited linear ubiquitination of RIP2 and NFκB activation in response to MDP, whereas knock-down of OTULIN had the opposite effect [185].

Importantly, OTULIN binds also to the catalytic subunit of LUBAC HOIP and restricts its autoubiquitination with linear ubiquitin chains, preventing activation of LUBAC under resting conditions [185]. However, a later study suggested that OTULIN does not form part of the NOD-SC and affects only LUBAC, but not RIP2 ubiquitination [189]. Instead, CYLD was shown to interact with the Peptide:N-glycanase/ubiquitin-associated (UBA) or ubiquitin X (UBX)-containing proteins (PUB) domain of HOIP like OTULIN, being recruited to the NOD-SC [190]. Similarly, CYLD is recruited by HOIP to the TNF Receptor Signaling Complex (TNFR-SC), where it restricts M1-linked ubiquitination, negatively regulating NFκB activation downstream of TNF signaling [189,191].

Contrary to OTULIN, CYLD does not affect LUBAC ubiquitination but antagonizes M1 and K63-linked ubiquitination of RIP2 [190]. Ablation of Cyld in different cell lines and BMDMs increased NFκB-target gene expression in response to MDP and infection with Listeria monocytogenes, resulting from increased M1 and K63-linked ubiquitination of RIP2 [190,192].

Recently, the JAMM DUB family member MYSM1 was proposed as a novel negative regulator of RIP2 [193]. Mysm1−/− BMDMs exhibited greater cytokine production upon stimulation with NOD1/2 ligands compared to WT cells, and similarly, Mysm1-deficient mice generated greater inflammatory output in response to MDP administration [193]. MYSM1 is recruited to ubiquitinated RIP2 and antagonizes heterotypic K27, K63, and M1-linked ubiquitin chains conjugated to K209 of RIP2, thereby attenuating NOD-SC assembly and downstream signaling [193].

### 6.2. Factors Regulating NOD1/2 Protein Levels

Upon stimulation with MDP, NOD2 is ubiquitinated and rapidly degraded in a proteasome-dependent manner in several cell lines and primary macrophages [194]. NOD2 was proposed to be associated with HSP90, which protects it from K48-linked ubiquitination. The authors identified Suppressor of Cytokine Signaling 3 (SOCS3), which interacts with NOD2 in overexpression experiments, and showed that silencing Socs3 expression partially inhibited MDP-induced NOD2-degradation [194]. A plausible scenario entails displacement of HSP90 during NOD2 activation, providing access to a yet unknown E3 ligase for NOD2 ubiquitination, although this will need further investigation.

The RING E3 ligase Zinc and Ring Finger 4 (ZNRF4) has been reported to inhibit NOD2 signaling [195]. ZNRF4 knock-down in human macrophages increased pro-inflammatory cytokine output upon MDP stimulation [195]. Mechanistically, this is achieved by ZNRF4-RIP2 interaction, RIP2 K48 poly-ubiquitination, and its subsequently proteasomal degradation [195]. ZNRF4 protein levels were rapidly upregulated upon MDP stimulation, and the interaction between ZNRF4 and RIP2 occurred upon MDP treatment [195]. This suggests that ZNRF4 marks RIP2 for degradation in an activation-dependent manner, although further mechanistic details will need to be determined in future studies.

Besides NOD1/2, over the last years, several NLR proteins have been found to modulate innate immune signaling pathways including TLR, RLR, and cGAS signaling (reviewed in [174]) and to affect activity and stability of several key signaling components. In this context, NLRP12 was reported to promote K48-linked ubiquitination and degradation of NOD2 in monocytes in response to MDP [194], as well as reducing TLR- and TNF-dependent NFκB activation [196]. The importance of NLRP12 is underscored by the findings of several mutations in NLRP12 associated with autoimmune and inflammatory disorders [197].

## 7. Negative Regulation of TNF Signaling

TNF signaling can generate two mutually exclusive different cellular outputs: on one side it can direct an anti-cell death pathway driven by the induction of pro-inflammatory cytokines via complex I. On the other side it can induce cell death through cytosolic complex II (Figure 2) [198].

Ligand binding to the TNF Receptor Superfamily Member 1A (TNFRSF1A/TNFR1) induces formation of a signaling complex at the plasma membrane composed of TNFRSF1A Associated Via Death Domain (TRADD), TRAF2, TRAF5, Receptor Interacting Serine/Threonine Kinase 1 (RIP1; a.k.a. RIPK1) and cIAP1/2, termed complex I (Figure 2). RIP1 is ubiquitinated with K63-linked ubiquitin chains by cIAP1/2, which are required to recruit the TAK1/TAB2/3 complex [199,200]. Moreover, ubiquitination of RIP1 with linear ubiquitin chains by LUBAC is required to recruit the NEMO/IKKα/IKKβ complex, which is ubiquitinated itself by LUBAC with linear ubiquitin chains (Figure 2) [130,201,202]. TAK1 activates IKKβ, which in turn phosphorylates IκBα, the inhibitor of NFκB, promoting its ubiquitination and degradation by the ubiquitin ligase complex SCF^βTrCP^. This enables the transcription factor NFκB to migrate to the nucleus, resulting in the transcription of proinflammatory cytokines [203].

Altogether, ubiquitination plays an important role in TNF-induced NFκB activation, and expression of pro-inflammatory cytokines and anti-apoptotic proteins. Lack of linear ubiquitin chains synthesized by LUBAC, or inhibition of cIAP1/2 results in impaired recruitment of several components of the TNFR1 signaling complex, and subsequent destabilization of it, directing the formation of a cell-death promoting secondary signaling complex [204,205].

The importance of linear ubiquitin chains in innate immune signaling is underscored by the fact that patients with mutations in the catalytic component of LUBAC, HOIP, suffer from severe immunological disorders [35,206]. The DUBs A20, CYLD and OTULIN have been implicated as important negative regulators of the TNFR1 complex, negatively regulating transcription of NFκB target genes, but also controlling TNF-induced cell death (reviewed in [136]).

### 7.1. Negative Regulation of RIP1

#### 7.1.1. A20

The A20 protein is encoded by the *Tnfaip3* gene, and it harbors both DUB activity and E3 ligase activity [207]. A20 is strongly induced upon TNF signaling, and potently inhibits NFκB signaling [208]. *Tnfaip3*-deficient mice suffer from spontaneous inflammation, cachexia and die prematurely, due to the inability of *Tnfaip3*-deficient cells to terminate TNF-induced NFκB activity [209]. Similarly, loss-of-function mutations of *Tnfaip3* in human results in systemic inflammation concomitant with increased proinflammatory cytokine expression [210]. Consistently, TNF-stimulated PBMCs and fibroblasts from patients with *Tnfaip3*-mutations had increased levels of K63-ubiquitinated RIP1 and NEMO [210]. Initially, it was proposed that A20 terminates TNF-induced NFκB activation by cleaving K63-ubiquitin from RIP1 via its N-terminal OTU domain and subsequently promoting its degradation by K48-linked ubiquitination via its zinc finger 4 (ZNF4) domain, thereby preventing cell death [207].

However, in vivo studies with mice expressing A20 lacking the catalytic C103 residue in the OTU domain, or a mutant ZNF4 domain missing E3 ligase activity (A20^OTU/OTU^ and A20^ZNF4/ZNF4^), showed that genetic ablation of a single domain caused only little spontaneous disease, not resembling the phenotype of *Tnfaip3^−/−^* mice [211,212,213]. This indicated that neither the ZNF4 nor the OTU domain are exclusively responsible for the function of A20. Moreover, cells derived from A20^OTU/OTU^ mice had no change or only slightly enhanced NFκB gene expression, unlike *Tnfaip3*-deficient cells. However, both A20^OTU/OTU^ and A20^ZNF4/ZNF4^ cells had increased levels of K48 and K63 ubiquitination RIP1 [211,212,213].

In addition, A20 was shown to have a cleavage preference towards K48 ubiquitin-linkage chains in vitro [214], although a recent study proposed that phosphorylated A20 was capable of cleaving K63-linked ubiquitin chains as well as in vivo [213]. Altogether, these results challenge the sole involvement of the DUB activity of A20 in the negative regulation of NFκB signaling, and indicate that an additional mechanism other than its catalytic activity might be involved, and that its DUB activity may be functionally redundant with other DUBs.

It was suggested that the function of A20 is dependent on the scaffolding protein TAX1BP1, which recruits the E3 ligase ITCH to the A20 ubiquitin editing complex and promotes the degradation of RIP1 [215]. *Itch*-deficient mice suffer from sever immunological defects and die prematurely, likely due to enhanced TNF-mediated NFκB activation [215]. Similarly, *Taxbp1^−/−^* mice die prematurely and are hypersensitive to low doses of TNFα and IL1β, due to increased NFκB activation [111]. However, it remains unclear how ITCH mechanistically promotes A20-mediated RIP1-ubiquitination [215].

Lastly, RNF11 was proposed to be part of the A20 ubiquitin-editing protein complex and required for TNF- and LPS-dependent K48-ubiquitination of RIP1 and TRAF6 [216]. Knock-down of *Rnf11* in monocytes increased activation of NFκB signaling in response to TNF and LPS, consistent with increased RIP1 and TRAF6 K63-linked ubiquitination [216].

#### 7.1.2. CYLD

CYLD is another important negative regulator of NFκB signaling. Patients affected from CYLD truncations suffer from cylindromatosis, a disease characterized by the formation of multiple tumors in the skin [217]. Similarly, *Cyld*-deficient mice are more susceptible to colitis-induced tumorigenesis and chemically-induced skin tumors due to increased NFκB activation [218,219].

Loss of *Cyld* expression was shown to enhance NFκB activation both in unstimulated conditions and in response to TNF and CD40 ligands [220,221], caused by the increase in M1- and K63-linked ubiquitination of components of the TNFR1 signaling complex [189,218].

CYLD is recruited to the TNFR1 complex by HOIP depending on the adapter protein Spermatogenesis Associated 2 (SPATA2), which was recently discovered to be crucial for CYLD recruitment and activation [222]. Upon recruitment to the TNFR1 complex, CYLD negatively regulates NFκB activation by removing M1-linked and K63-linked ubiquitin chains from several components of the TNFR1 signaling complex [189,218,220,221,223,224]. Knock-out of CYLD resulted in increased M1 and K63 ubiquitination of RIP1, TNFR1, and TRADD [189,225,226].

#### 7.1.3. LUBAC and OTULIN

A third DUB, OTULIN has been shown to be involved in TNF-mediated NFκB activation and cell death. OTULIN is the only DUB that has been identified so far that cleaves exclusively M1-linked ubiquitin chains [188]. Mutations in *Otulin* were found to be embryonic lethal and *Otulin*-deficiency in immune cells resulted in overproduction of cytokines and inflammation due to increased levels of M1-ubiquitination [188]. Similarly, patients with a homozygous *Otulin* mutations suffer from acute autoinflammation [227,228]. OTULIN interacts with HOIP through its PUB-interacting motif (PIM), antagonizing M1-linked ubiquitin chains synthesized by LUBAC [188,189,191]. Thus, while the DUB OTULIN is part of the LUBAC complex, and counters its synthesized M1 poly-ubiquitin chains [188,189,191], how the balance between these opposing biological output is controlled remains largely unknown.

### 7.2. Negative Regulation of TRAFs in the NFκB Pathway by DUBs and E3 Ligases

TRAF proteins are important players in the regulation of signal transduction of various immune signaling pathways, regulating activation of type I IFN-, NFκB-, and MAPK-signaling. TRAFs are recruited to several signaling adapters and are frequently subjected to K63-linked ubiquitination, acting as scaffold for the recruitment of downstream kinases TBK1/IKKε or IKKα/IKKβ/NEMO (Figure 2) [44,45]. Therefore, it is not surprising, that their activity is tightly regulated by several E3 ligases and DUBs.

#### 7.2.1. CYLD

CYLD was reported to target K63-linked ubiquitin chains of TRAF6, TRAF2, and NEMO in response to TNF and IL-2 stimulation [220,221,223,229]. Moreover, ectopic expression of CYLD negatively regulates MAPK signaling and NFκB activation in response to various TLR-ligands by inhibiting TRAF6 and TRAF7 ubiquitination, which is markedly increased by CYLD knock-down [230].

#### 7.2.2. A20

The DUB A20 terminates TLR-induced NFκB by removing K63-linked ubiquitin chains from TRAF6 [231]. Furthermore, A20 was has been reported to inhibit also TRIF-dependent NFκB, but not IRF3 activation, by restricting TRAF6 ubiquitination [232]. Together with the regulatory protein TAX1BP1, A20 was shown to inhibit the E3 ligase activity of TRAF6, TRAF2, and cIAP1, by antagonizing their interaction with the E2 enzymes Ubiquitin Conjugating Enzyme E2 N (UBE2N) and UBE2D3 and promoting their proteasomal degradation, downstream of both TNF and TLR signaling [233]. However, it is not clear whether A20 sterically interferes with the interaction between these E2 enzymes and TRAF6, TRAF2, and cIAP1, or whether A20 directly modifies them and thereby antagonizes their interaction [233].

#### 7.2.3. Zinc Finger CCCH-Type Containing 12A (ZC3H12A)

The ribonuclease ZC3H12A (a.k.a. MCPIP1, and Regnase-1), is a well-recognized suppressor of the inflammatory response. It is induced by inflammatory cytokines, and binds Untranslated Regions (UTR) of various mRNAs encoding inflammatory cytokines, and induces their degradation [234]. ZC3H12A was identified as a negative regulator of TRAFs by acting as a DUB [235]. *Zc3h12a* deficiency resulted in constitutive ubiquitination of TRAF2, TRAF3, and TRAF6 in LPS-stimulated splenocytes [235]. This translated into premature death of mice, caused by increased levels of TNF, IL1β, IL6, or MCP-1 in macrophages under normal and LPS-stimulated conditions [235]. ZC3H12A was proposed to harbor a novel DUB domain and a putative UBA domain, allowing the interaction with ubiquitinated substrates [235]. However, given that ZC3H12A was able to cleave K48 and K63 linked ubiquitin chains in vitro, and affected the global ubiquitination level when overexpressed in cells, it might also have other targets, or be involved in other processes [235].

#### 7.2.4. FBXL2

F-Box and Leucine Rich Repeat Protein 2 (FBXL2) was identified as a sentinel TRAF-inhibitor in epithelia and monocytes, by mediating their ubiquitination and degradation, as such negatively regulating cytokine expression [236]. The activity of the SCF^FBXL2^ complex was shown to be regulated by the E3 ligase F-Box Protein 3 (FBXO3), which ubiquitinates FBXL2 [236]. Importantly, recruitment of FBXO3 was shown to be facilitated through FBXL2 phosphorylation by Glycogen Synthase Kinase 3 Beta (GSK3β), creating a phospho-degron [236].

The authors showed that ectopic expression of FBXO3 in mice resulted in increased cytokine levels in the lung and increased mortality upon infection with *Pseudomonas aeruginosa*, compared to WT mice, or mice expressing a loss-of function polymorphism variant FBXO3^V221I^ occurring in European Caucasians. In line with inflammation enhancement by FBXO3, its silencing ameliorated *Pseudomonas aeruginosa* induced lung injury [236]. Similarly, patients with septic shock displayed and increase in FBXO3 and TRAF protein levels, concomitant with a decrease in FBXL2 levels [236].

The importance of FBXL2 and FBXO3 was underscored by two other studies, as targeting FBXO3 in rodents with the small inhibitor BC-1215 significantly ameliorated H1N1-influenza induced lung injury and acute lung injury, by increasing FBLX2 protein levels and decreasing TRAF protein levels [237,238].

#### 7.2.5. RNF19A

The RING-type E3 ligase RNF19A has been reported to promote degradation of TRAF6 by degradative ubiquitination at K356, K365 and K371 [239]. RNF19A was shown to be recruited to TRAF6 by the NOD family member NLRP11, which is rapidly induced by LPS in macrophages and then rapidly degraded itself, suggesting that it might act specifically during the resolution phase of infection [239]. However, NLRP11 reduces TRAF6 protein levels also under steady-state conditions, likely contributing to basal TRAF6 turnover. Knock-out of *Rnf19a* or *Nlrp11* increased TLR-ligand induced production of TNF, IL6, and IL1β in THP1 cells and peripheral blood mononuclear cells [239].

#### 7.2.6. TRIM38

Silencing *Trim38* expression in vivo by intraperitoneally administered siRNAs increased TNFα and IL6 expression in macrophages upon LPS, poly(I:C), and lipoteichoic acid stimulation [240]. Further overexpression studies showed that TRIM38 interacts with TRAF6 via its C-terminal PRY/SPRY domain, and targets TRAF6 with K48-linked ubiquitin chains for proteasomal degradation [240]. Of note, the authors showed that TRAF6 promotes also K63-linked ubiquitination of TRAF6 [240], which would suggest a positive regulatory role of TRIM38 in regards to TRAF6, and support the results from a previous study that identified TRIM38 in a luciferase reporter gene screen for cDNAs that positively regulate NFκB and MAPK signaling [241].

### 7.3. Negative Regulation of the TAK1/TAB1/2/3 and IKKα/β/NEMO Complexes

#### 7.3.1. ITCH

Sustained activity of the kinase TAK1 was shown to be inhibited by a complex formed by the DUB CYLD and E3 ligase ITCH in response to TNF signaling [224]. In a sequential mechanism, CYLD removes K63-linked ubiquitin chains from TAK1, which are then replaced by K48-linked ubiquitin chains by ITCH, however the targeted lysine residues remain undefined. BMDMs from *Cyld^−/−^* and *Itch^−/−^* mice had sustained activation of TAK1 and chronic production of inflammatory cytokines upon stimulation with TNF, which was rescued by inhibiting TAK1 activity with the drug (5Z)-7-Oxozeaenol or by rescuing CYLD and ITCH expression [224]. Subsequent studies indicated that K72 of TAK1 might be critical for TNF-induced K48-linked ubiquitination of TAK1 by ITCH [242,243]. Of note, CYLD and ITCH complex formation occurred specifically after TNF stimulation, resulting in a negative feed-back loop to downregulate NFκB activation [224].

#### 7.3.2. TRIM29

The E3 ligase TRIM29, which is highly expressed in lung tissue [169,244] and was shown to restrict type I IFN production by targeting MAVS and STING [104,105,169], has also been identified as a negative regulator of NEMO [244]. Alveolar macrophages from *Trim29^−/−^* mice were reported to have increased production of IFNα, IFNβ, TNF, and IL6 compared to WT cells in response to RLR ligands. In line with this notion, *Trim29^−/−^* mice were less susceptible to infection with influenza A virus and reovirus, with lower viral loads in the lungs [244]. In contrast, *Trim29^−/−^* mice were more susceptible to intranasal LPS challenge or infection with *H. influenzae* compared to wild type mice, caused by septic shock due to overproduction of cytokines. TRIM29 was shown to limit the abundance of NEMO under basal conditions, however, infection with reovirus, but not LPS, further upregulated TRIM29 levels promoting proteasomal degradation of NEMO by ubiquitination at K183 [244].

## 8. Negative Regulation of IFN Signaling and Induced Response Proteins

Type I (IFNα, and IFNβ), type II (IFNγ) and type III (IFNλ) IFNs are cytokines produced during infections, some other cellular stress conditions, and upon aberrant recognition of ‘self’ molecules in auto-immune disorders collectively known as interferonopathies [31,245,246,247]. Although other type I IFNs exist with poorly understood functions (IFNδ, IFNε, IFNζ, IFNκ, IFNτ, IFNω), IFNα and IFNβ are particularly recognized for their essential role in anti-viral restriction [248]. This is underpinned by the fact that mice lacking the Type I IFN Receptor (IFNAR1), are hypersusceptible to viral infections [249]. IFNγ plays a key role in anti-bacterial defense, for an important part by initiating signaling on macrophages through the IFNγ receptor [250,251,252]. Moreover, IFNγ contributes to the direct antiviral response, as well as indirectly through stimulation of adaptive immunity (reviewed in [253]).

Recognition of these IFNs by their specific receptors activates kinases: Janus Kinase 1 (JAK1) and Tyrosine Kinase 2 (TYK2) for type I IFNs, and JAK1 and JAK2 for IFNγ (Figure 3) [245,246,247]. These, in turn, phosphorylate inactive cytoplasmic transcription factors, which ultimately translocate to the nucleus where they drive the expression of hundreds of ISGs with direct anti-pathogen activities, as well as immune-modulatory factors [254,255]. Type I IFN signaling drives phosphorylation of STAT1 and STAT2, which associate with the DNA-binding unit IRF9 into a hetero-trimeric TF complex named Interferon Stimulated Gene Factor 3 (ISGF3) [245,246]. IFNγ signaling on the other hand, results in STAT1 phosphorylation, and its subsequent homodimerization [245,246].

All of the core proteins of these JAK-STAT pathways are constitutively present in non-activated cells [256,257,258]. Under non-stimulated conditions these JAK-STAT pathway proteins are considered stable, although this has not been systematically investigated to our knowledge. Their activation and inactivation are predominantly regulated post-translationally by phosphorylation [256,257,258]. However, additional cellular mechanisms controlling their cellular concentrations are in place, ensuring their proper protein levels post-activation [245,246]. This contributes to timed-deactivation of the innate immune response, and is moreover significant as some of the JAK-STAT proteins are themselves ISGs, and thus upregulated through their own activity [255,259].

Studies with *Stat1*, *Stat2*, and *Irf9* knock-out mice and their derived cells, showed that ablation of each of these components drastically decreased the basal expression of all three ISGF3 components [259]. From this it has become clear that basal STAT1, STAT2, and IRF9 protein synthesis is importantly determined by the rate of self-transcription by the ISGF3 complex [259]. However, the factors that on the flip-side turn-over non-activated JAKs and STATs, thereby contributing to establishment of a steady-state protein equilibrium, have remained unidentified. It goes beyond the scope of this review to discuss all individual reported regulators of the type I and II IFN systems. A comprehensive overview can be found in Table 1, as well as various other reviews [31,260,261,262]. Below, we discuss several regulatory mechanisms of the IFN-related proteome during non-stimulated conditions, as well as immune-resolution, which exemplify some of the fundamental processes at play to control innate immune dynamics.

### 8.1. IFN Receptors

In contrast to regulation of basal JAK-STAT pathway protein levels, several cellular protein turnover mechanisms have been discovered that limit accumulation specifically of the activated forms. IFNAR1 itself is turned-over in a ligand-induced state [263]. Studies by Serge Fuchs’s group identified a binding site in the IFNAR1 cytoplasmic tail for the SCF^βTrCP^ E3 ligase complex, which results in IFNAR1 ubiquitination, and subsequent lysosomal degradation [263].

Interestingly, Beta-Transducin Repeat Containing (βTrCP) is a well-known Cullin substrate adaptor recognizing phospho-degrons. Its best-known example being a phosphorylated peptide in the NFκB inhibitor (IκBα), which it targets for proteasomal degradation [264]. The IFNAR1 cytoplasmic tail has a similar peptide [263], which is phosphorylated upon ligand binding, and subsequently recognized by βTrCP for ubiquitination on a nearby lysine [263]. The stress kinase p38 has been identified as one of the possible mediators of IFNAR1 phosphorylation [265], which may in part explain specific IFNAR1 targeting upon activation. The IFNγ receptor is likewise degraded after targeting to the lysosome, which was shown to be dependent on phosphorylation by glycogen synthase kinase 3 beta (GSK3β) [266], although whether this activity is coupled to immune-activation remains undetermined.

### 8.2. Janus Kinases

Downstream of the IFN receptors, some of the JAK and STAT protein levels are down-regulated through proteasomal degradation upon pathway activation. One family of well-recognized negative feedback inhibitors are the Suppressor of Cytokine Signaling (SOCS) proteins, substrate adaptors for Cullin E3 ligases [267,268]. In particular SOCS1 is recognized to play a crucial role in curtailing type I and II IFN signaling [267,268]. Its importance is exemplified by the fact that *Socs1* deficiency is neonatally lethal in mice, which results from severe IFNγ-dependent hyper-inflammation [267]. In line with this notion, *Socs1^−/−^*/*Ifng*^−/−^ double-knock-out rescued neonatal lethality in mice, although they still developed inflammatory disease later on pointing at additional SOCS1 targets which may be IFNγ-signaling independent [267].

SOCS1 is an ISG and as such itself induced by type I and II IFN signaling, creating a negative feedback loop upon prolonged receptor stimulation [267]. Structural and biochemical analysis revealed that mechanistically SOCS1 inhibits JAK-dependent signaling in two complementary manners. Firstly, SOCS1 can directly bind the JAK1, JAK2, and TYK2 kinase domains, thereby effectively preventing substrate binding and kinase function [268]. This interaction is specific for the unphosphorylated (i.e., non-activated) JAK1 [268], suggesting that SOCS proteins lock these JAK kinases in inactive forms, and as such shift the cellular equilibrium to an off-state.

In addition, it has been long recognized that SOCS proteins act as substrate receptors for the Cullin5/ElonginBC E3 ligase complex [269]. In that context, SOCS1 has been shown to mark JAK2 for proteasomal degradation in cell-based assays, by binding its active phosphorylated form, and subsequently ubiquitinating it [269]. This may point to SOCS1 specifically targeting the activate JAK pool for degradation, while maintaining an inactive JAK protein pool for activation post-resolution.

In line with this notion, differentiation of BMDMs to osteoclasts by the cytokine Receptor Activator Of Nuclear Factor Kappa B Ligand (RANKL), is likewise accompanied by proteasomal JAK1 degradation, thereby releasing IFNβ-dependent inhibition of this differentiation process [270]. Whether this process is SOCS1-dependent remains thus far undetermined. Importantly, this indicated that regulation of JAK protein levels may be cell-type specific. This is further substantiated by the report that the E3 ligase CBL specifically degrades activated JAK2 in hematopoietic stems cells [271], in which its over-activation has been identified as an important oncogenic driver.

Together, these findings indicate that JAK kinases are critically controlled in their activated state by proteasomal degradation. However, individual cell types may employ different mechanisms to prevent over-activation, some of which may still remain to be discovered.

### 8.3. STAT Transcription Factors

The protein levels of STAT1, STAT2, and IRF9 are constitutively maintained [259]. However, *Stat1* is an ISG, and consequently induces its own protein levels during IFN-signaling [272]. Consequently, cellular mechanisms are in place to degrade this excess pool of STAT1 during resolution, and bring back the protein equilibrium to pre-activation levels.

Following IFNγ stimulation, phosphorylation of S727 is key for STAT1-driven response gene transcription. An initial study by Kim and Maniatis discovered that treatment of IFNγ-stimulated HeLa cells with proteasome inhibitor, specifically stabilized phosphorylated STAT1 [272]. This indicated the existence of cellular mechanisms specifically targeting the activated STAT1 homodimer pool for proteasomal degradation. In line with this notion, an S727A STAT1 mutant was not ubiquitinated under these conditions [272].

Importantly, most cells express two STAT1 isoforms. The larger STAT1α isoform can be phosphorylated on S727, and is considered the major transcriptional driver of IFNγ-induced genes. In contrast, the C-terminally truncated STAT1β isoform lacks the S727 residue, and is considered a competitive negative regulator. In line with a critical role of S727 phosphorylation-dependent STAT1 degradation, studies in esophageal squamous cell carcinoma cells found that STAT1β association with STAT1α prevented STAT1α proteasomal degradation [273]. However, later work from the same group in the same cell type indicated ERK inhibitors decreased STAT1 turnover, yet unexpectedly neither an S727A, nor a Y701F mutant were more stable than their WT counterparts [274].

Many of the above-described conclusions drawn on phosphorylation-requirement for degradation are based on over-expressed mutants [273,274], which may substantially influence protein stability, or skew observations based on the expression of a single isoform. Therefore, future studies using genome-editing to mutate phospho-sites in the endogenous *Stat1* locus in various different cell types may be required to get a better insight into activation-dependent STAT1 turnover mechanisms.

Similar to what has been reported for IFNγ, stimulation with type I IFNs is also accompanied by ubiquitination and proteasomal degradation of STAT1 [275], which likewise is selective for Y701-phosphorylated, activated nuclear STAT1. Interestingly, the authors discovered that the DUB USP2a is rapidly imported into the nucleus upon type I IFN stimulation, deubiquitinates STAT1, thereby upregulating nuclear STAT1 levels, and ISG transcription [275].

Thus far, most reported cellular degradation pathways point to specifically targeting the activated STAT1 pools. However, mass-spectrometry analysis of proteasome-associated proteins previously identified STAT1α to be associated under non-activated conditions [276]. This could suggest that additional STAT1 proteasomal targeting mechanisms may be at play for homeostatic STAT1 turnover, although whether this substantially regulates steady-state STAT1 protein concentrations remains to be determined.

Taken together, these findings underpin the importance of cellular degradation mechanisms specifically targeting activated STAT1 complexes during type I and type II signaling. However, whether additional pathways mediate slower homeostatic turnover of inactive STAT1 will require further investigation.

### 8.4. Degradation of Innate Immune Response Proteins

Type I and II IFN stimulation transcriptionally induces 300–600 response genes [254,255]. While the protein level increases for all these factors have not been systematically characterized, a significant fraction is undetectable in non-stimulated conditions, yet highly expressed post-stimulation [254,255]. Most of the well-studied ISGs—such as Protein Kinase R (PKR), 2’-5’-Oligoadenylate Synthetase 1 (OAS1), IRF7, and RNAseL—require activation by infection-dependent signals, whereas select other ISGs—such as IRF1—are synthesized in a constitutively active form [10,277,278,279].

This raises the question of how innate immune proteins are targeted for degradation, how this is regulated for constitutively expressed proteins compared to their induced counterparts, and whether there are fundamental differences in how cells target these types of proteins for proteasomal degradation.

It goes beyond the scope of this review to discuss the described proteasomal degradation mechanism of all response factors. Detailed reviews on degradation mechanism for ISGs can be found in Table 1. Below we describe degradation mechanisms for some of the key members of the IRF family. The three discussed IRF family members represent three conceptual types of protein regulation, exemplifying: (**i**) stably maintained, non-induced (IRF3); (**ii**) activation-induced degradation (IRF7); as well as (**iii**) degradation-by-default (IRF1) principles for determining concentrations of induced response proteins.

IRFs are key immune transcription factors, with different family members driving both innate, as well as adaptive immune processes [245,246]. IRF3 is constitutively expressed in virtually all nucleated cell types. Its phosphorylation, and subsequent nuclear translocation is triggered post pathogen recognition, and drives transcription of IFNβ [280].

IRF7 is a closely related family member, which also requires pathogen-activated phosphorylation, and is responsible for IFNα transcription [31]. In contrast to IRF3, IRF7 is not or lowly expressed in most non-stimulated cell types, with plasmacytoid dendritic cells (pDC) being a notable exception, in which it is expressed at high constitutive concentrations [281]. IRF7 is an ISG, and thus induced upon IFNβ signaling, priming exposed cells for rapid IFNα synthesis upon pathogen exposure [281].

Conceptually, this implies that IRF3 protein levels are predominantly maintained at constant homeostatic levels, and its output is mainly controlled by phosphorylation and dephosphorylation [280,282]. While various mediators of IRF3 degradation have been described (see Table 1), many of them have been reported to target multiple IRFs, and appear to do so independently of activation. This suggests that while their activity may indeed influence IRF3 protein concentrations, it could predominantly establish a constant steady-state IRF3 protein content, and represent general effects of protein quality control turnover of nascent IRF3 proteins. The notable exception is Peptidyl-Prolyl Cis-Trans Isomerase Never In Mitosis (NIMA)-Interacting (PIN1), which was shown to interact with the IRF3 dimer upon RNA virus-dependent phosphorylation of a serine-proline motif and promote its degradation in the nucleus by an unknown E3 ligase [283].

Indeed, E3 ligases such as Ubiquitin Protein Ligase E3C (UBE3C/RAUL) described to mediate IRF3 degradation [8], are well recognized as part of the general proteasome-associated degradation machinery from humans to yeast [284,285,286]. UBE3C was shown to regulate WT and phosphorylation deficient IRF3 and IRF7 levels in homeostatic conditions, as well as during infection, suggesting that it might target the IRFs during steady state conditions as a part of protein QC, and independently of their activation [8]. However, it remains to be determined whether increased levels of the IRFs upon *Ube3c* ablation result from additional changes in proteasome processivity causing partial degradation of these IRFs, after being targeted to the proteasome by an additional E3 ligase [285].

In contrast, IRF7 protein requires active turnover during immune resolution, for returning to its low or absent homeostatic levels. Importantly, this suggests that pDCs employ specialized strategies to ensure high constitutive IRF7 protein concentrations. Indeed, IRF7 is degraded through proteasomal degradation, and has been reported to have a half-life of 30–60 min [287,288]. Experiments with phosphorylation-deficient and phospho-mimetic mutants indicated that while part of the IRF7 pool can be degraded in a phospho-independent manner, its activation does contribute to increased IRF7 turnover [288].

Importantly, IRF7 turnover is cell-type dependent; for example, IRF7 degradation rates are significantly lower in splenocytes and thymocytes, compared to its half-life in MEFs [288], suggesting that parts of the IRF7 degradation machinery may be differentially expressed or activated in different cell types.

Moreover, the specialized IFNα-producing pDCs maintain high levels of IRF7 [281]. Mechanistically, TRIM8 has been reported to bind phosphorylated IRF7, and prevent in an E3 ligase-independent manner, IRF7 association with PIN1 [283]. PIN1 recognizes phosphorylated proline motifs and catalyzes peptide bond isomerization, thereby affecting their fate [283]. While for IRF3 this has been described to confer ubiquitination and degradation [283], the authors concluded that TRIM8 competes with PIN1 for IRF7 binding, thereby stabilizing it without major effects on its ubiquitination status [289]. The detailed mechanism of PIN1-dependent IRF7 stability regulation has remained unclear.

Despite this, and other putatively unknown mechanisms ensuring high steady-state IRF7 levels in pDCs, there is accumulating evidence that IRF7 is specifically targeted for degradation upon activation in this cell type, and possibly other myeloid cells [9,290]. In this context, ablation of *Trim35* prevented K48 ubiquitination of IRF7 in macrophages and pDCs [290], suggesting that TRIM35 ubiquitinates and degrades IRF7. *Trim35* mRNA levels are relatively high in mouse macrophages and DCs, and can be induced upon stimulation [291,292]. This may explain the cell type-specific regulation by TRIM35, although the exact nature of TRIM35 activation and being part of a feedback loop remain to be clarified. In line with the notion that even in myeloid cells with higher steady-state levels, negative feedback loops resulting in IRF7 degradation are in place, immune-induced SOCS1 has been reported to target IRF7 in TLR-activated human pDCs [9].

Together, these findings emphasize that both activation-induced, as well as basal degradation mechanisms are contributing to IRF7 concentrations. Further research will be needed to unravel how these mechanisms act in unison to ensure viable homeostatic conditions, as well as immune dynamics.

In contrast to IRF3 and IRF7, IRF1 is synthesized in a constitutively active form [279,293,294]. IRF1 transcription is induced by both type I and II IFNs in many different cell types [279,293,294]. While IRF1 drives the transcription of various immune response genes, similarly to its IRF3/7 counterparts, it additionally induces cell cycle inhibitory factors [295]. As such, it is considered an important tumor suppressor [296,297]. Indeed, *Irf1^−/−^* mice have a loss of antitumor functions, their cells are more susceptible to cell transformation, and fail to induce cell cycle arrest upon DNA damage [296,297].

Like its non-canonical always-on functional nature, the fate of the IRF1 protein itself is also unlike IRF3/7. To date there is accumulating evidence that IRF1 is degraded-by-default, and its steady-state protein levels are predominantly determined by the equilibrium between its transcriptionally-induced synthesis, and its constant rapid turnover [10].

IRF1 half-life was determined to be between 20 and 30 min [10]. Subsequent mutagenesis analysis revealed that the C-terminal domain is a main determinant of this instability [10]. The ubiquitin E3 ligase STUB1/CHIP has been identified as one of the factors targeting IRF1 in this context [277]. STUB1 is an E3 ligase implicated in general protein QC, and degradation of proteins with disordered domains. As such, it has been suggested that STUB1 recognizes the disordered IRF1 C-terminus, mediating its degradation by default [277]. Importantly, HSP90 has been reported to bind the IRF1 C-terminus as well, and have a stabilizing function [278], possibly by competitively preventing STUB1 binding. In line with this notion, treatment with HSP90 inhibitors further destabilized IRF1 [278].

IRF1 is phosphorylated at multiple residues: constitutively by casein kinase II, and in an immune-induced manner by IKKε [298,299]. Although the functional significance for its transcriptional activity may be rather limited, recent reports indicate that these sites may generate phospho-degrons that recruit hitherto unknown E3 ligases mediating IRF1 ubiquitination and degradation [298,299].

In line with this idea, the cancer associated S/T kinase GSK3β has been shown to phosphorylate IRF1 as well, generating a phospho-degron, recognized by the SCF^βTrCP^ Cullin E3 ligase complex [300]. Interestingly, GSK3β-dependent phosphorylation at T181 in the middle of the IRF1 protein was required for IRF1 transcriptional activation, as well as its degradation, indicating a putative activation-coupled turnover mechanism independent from the C-terminal destabilizing region [300].

Together, these mechanisms of IRF turnover exemplify cellular mechanisms for returning the immune-induced proteome to homeostatic conditions. It underpins the importance of differentiating between general protein QC and immune-specific degradation mechanisms.

## 9. Concluding Remarks

Scientific progress over the last decades has identified many negative regulators of the innate immune system. While some of them were identified as risk factors in hyper-immune disorders, many were discovered through genetic and protein-interaction screens. For a growing number of these factors, knock-out mouse models indeed firmly established their importance for innate immune control.

However, one of the arising messages from the works reviewed above, is that for many of these regulators it still remains unclear how they are themselves recruited, activated, and possibly degraded. A number of DUBs may be recruited to the protein complexes that synthesize the poly-ubiquitin chains that they target, raising the question whether there are general principles at play that determine their recruitment, activation, and activity. On the side of E3 ligases, in many instances their knock-out results in immune disorders in mouse models, while their regulation has remained likewise ill-defined. Future structural studies of active and inactive DUB confirmations, combined with biochemical and cell-biological analyses will be key for better understanding such regulatory mechanisms.

Some of the key families of negative regulators—such as the SOCS proteins—have been recognized for their crucial contributions in degrading immune proteins. They are transcriptionally induced during the immune response, providing an attractive negative feedback model. However, how they are themselves removed during resolution, whether they are co-degraded, or can bind new substrates after release and degradation of the previously bound ones, remain key questions for future studies.

For surprisingly few of the hundreds of transcriptionally induced innate immune response proteins it is clear how they are removed upon immune resolution. The few that have been described in some detail suggest that the presence of disordered degron sequences may be an often-occurring theme. As for IRF1, some of them could be primary binding sites or ubiquitination sites for E3s, and may thus be controlled by factors that compete with binding of disordered regions, such as HSPs and other chaperones.

On the other hand, disordered regions are also in principle accessible from a structural stand-point, and therefore excellent sites for phosphorylation. In particular, phospho-degrons have been long recognized and could be a common theme in response protein degradation [264], as is the case for IRF7. Given that numerous stress kinases are key for innate immune signaling, it is possible that these same factors put on the phospho-degron marks that degraded the targets they induce. Whether this is indeed a wider theme, will require more detailed studies.

In this context, an important step will be to identify all rapidly degraded response proteins during the immune response. On the one hand, stability analysis of ectopically expressed response proteins may provide key insights [301]. This may be an attractive approach, as the use of an exogenous promotor to express these proteins would allow screening for effects on protein concentrations, without confounding effects on response factor transcription. Moreover, emerging techniques to identify proteasome-engaged substrates by mass-spectrometry may provide effective alternative means to identify actively degraded immune response proteins upon infection or cytokine stimulation [302], although detection sensitivity may be a limiting factor for some lowly expressed response proteins.

The list of identified factors that dampen the innate immune response has steadily grown over the last years, and with the advances in genetic and proteasome screening, this number will undoubtedly continue to grow. Perhaps of equal importance will be to extend such efforts to identify factors that are required for these DUBs and E3 ligases to function in their cellular environment.

Validation of newly identified regulators in mouse models and the effect of their knockouts on immune over-activation will be crucial to establish their biological significance. However, as exemplified by many of the regulators described in this review, the cellular and molecular means by which these regulators modulate their substrates, whether they recognize specifically activated substrates, how they are themselves activated, and where in the cell they interact will be key to investigate further in the years to come. While cell biological models will be important for this, some of mechanistic insights to these regulatory mechanisms can likely be exclusively be achieved through biochemical and structural analyses.

**Table 1 viruses-13-00584-t001:** Negative regulators of innate immune signaling molecules.

Sensor Layer	E3 Ligase	DUB
RIG-I	RNF125 [74] CBL [75] RNF122 [76] STUB1/CHIP [77,78] TRIM40 [80] MARCH5 [79]	CYLD [67,68] USP14 [71] USP21 [70] USP27X [73] USP3 [69] USP25 [72]
MDA5	RNF125 [74] TRIM40 [80] TRIM13 [84]	USP21 [70] USP3 [69]
TRIM25	LUBAC [12]	
cGAS	Unknown [161]	
IFI16	TRIM21 [160]	
DDX41	TRIM21 [159]	
NOD2	TRIM27 [303] RNF34 [304] HSP90/SOCS3 [194]	
TLR3, TLR4, TLR5, TLR9	RNF216/TRIAD3A [123]	
TLR3	RNF170 [139]	
TNFR1		CYLD [189]
IFNAR1	βTrCP [263,265,305,306] Unknown [307,308,309,310,311,312,313]	
IFNGR1	STUB1/CHIP [314,315] Unknown [266]	
Adapter layer	E3 ligase	DUB
MAVS	RNF125 [74] SMURF1/2 [106,107] RNF5 [316] ITCH [97,102] MARCH5 [79,96] TRIM25 [103] TRIM29 [105]	OTUD3 [95] YOD1 [317] OTUD1 (indirectly) [108]
STING	TRIM29 [104,169] RNF5 [167] TRIM30α [170] RNF26 [168]	USP13 [171]
MYD88	NRDP1 [65] SMURF1 [318] SMURF2 [318] CBLB [140,141] SCF^SPOP^ [142,145,146]	OTUD4 [135] CYLD [137]
TRIF	RNF216/TRIAD3A [124] WWP2 [148] CBLB [141] TRIM38 [319,320]	A20 [321]
TIRAP	RNF216/TRIAD3A [124]	
TRADD		CYLD [189]
TRAF3	RNF216/TRIAD3A [125] SMURF1 [108,322] PARKIN [323] SCF^FBXL2^ [236] cIAP1/2 [324]	OTUD5 [119] OTUB1 [325] OTUB2 [325] UCHL1 [326] MYSM1 [122] USP25 [327,328] MCPIP [235]
TRAF6	TRIM38 [240] RNF19A [239] WWP1 [149] CBL [329] RNF11 [216]	USP25 [72] USP2a [330] OTUB1 [325] OTUB2 [325] UCHL1 [326] MYSM1 [122] A20 [216,231,232], (DUB act.-indep.) [233] CYLD [220,221,223,230] MCPIP [235] USP4 [331]
TRAF7		CYLD [230]
TRAF2	SIAH2 [332]	A20 (DUB act.-indep.) [233] CYLD [220,221,223] MCPIP [235] USP4 [331]
cIAP1		A20 (DUB act.-indep.) [233]
Kinase layer	E3 ligase	DUB
RIP1	RNF216/TRIAD3A [124] A20 [207,211,212,213] ITCH [215] RNF11 [216]	A20 [207,210,211,212] CYLD [189,225,226] OTUD7B [333] USP21 [334] USP4 [335] OTULIN [188]
RIP2	ZNRF4 [195] ITCH [336] (indirectly)	A20 [187] OTULIN [185] CYLD [190] MYSM1 [193]
LUBAC (E3 ligase)		OTULIN [188,189,191]
TBK1	DTX4 [113,114,115] TRAIP [116] TRIM27 [118] SOCS3 [337] TRIM11 [338] RNF144B [339] ASB8 [340]	CYLD [67] USP38 [117] USP2b [341] A20 [110,342]
TAK1		USP18 [343]
TAB2/3	TRIM30α [344]	
NEMO	TRIM29 [244] TRAF7 [345] TRIM40 [346] TRAF4 [347]	CYLD [218,220,223] A20 [210,348] USP18 [343] USP7 [349] UCHL1 [326] OTULIN [191]
TAK1	ITCH [224,242,243]	CYLD [224] USP4 [350] USP19 [351]
TAB1	ITCH [352]	
TAB2/3	HOIL-1 [353] TRIM38 [354] RNF4 [355]	
JAK1	RNF125 [87] Unknown [270,356]	
JAK2	CBL [271,357] SOCS1 [269,358] Cullin5/ElonginBC [269]	USP9X [359]
TYK2	SIAH2 [360] Unknown [361]	
Transcription factor layer	E3 ligase	DUB
IRF3	SCF complex [362] HOIL1/RBCK1 [363] TRIM21 [364,365] TRIM22 [364] PIN1+unknown E3 ligase [283] UBE3C/RAUL [8] TRIM26 [366] CBL [367] RNF26 [168]	OTUD1 [368] SENP2 [369]
STAT1	SMURF1 [370] PDLIM2 [371] Unknown [272,274,372,373,374,375]	USP2a [275] USP13 [376]
STAT2	DCST1 [377] FBXW7 [378]	
IκBα	SCF^βTrCP^ [379,380] TRIM22 [381]	
NFκB	PDLIM2 [382] SOCS1 [383] EloB/C/Cul2/SOCS1 [384,385] MKRN2 [386] RNF182 [387]	
Response protein layer	E3 ligase	DUB
IRF1	STUB1/CHIP [278,388] MDM2 [389] βTrCP [300]	
IRF7	SCF complex [288] TRIM21 [390] RAUL/UBE3C [8] SOCS1 [9] SOCS3 [9] NDRG1 [391] TRIM35+PIN1 [290] Unknown [288,299]	
RNaseL	Unknown [392,393]	
Viperin	UBE4A [394] Unknown [395]	
TTP	βTrCP [396] Ub-independent [397] Unknown [398,399,400,401]	

## Figures and Tables

**Figure 1 viruses-13-00584-f001:**
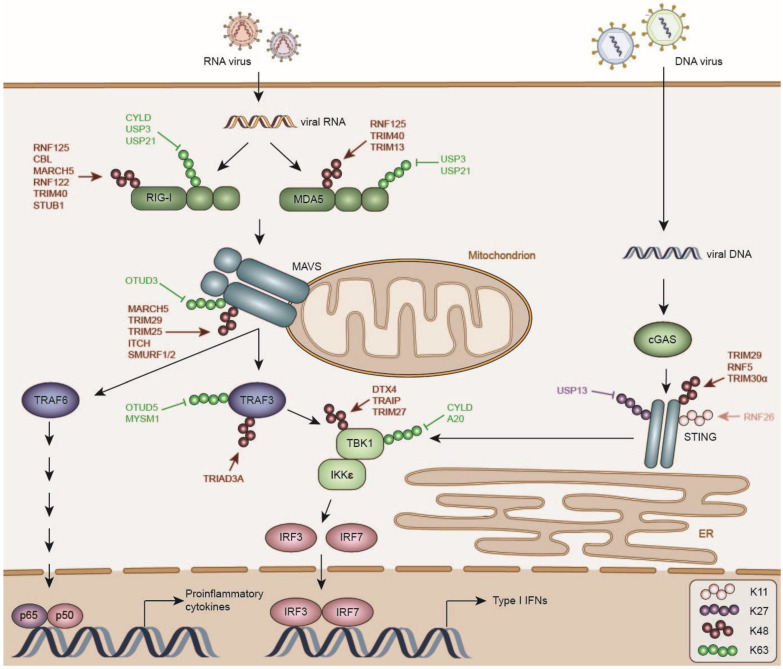
Negative regulation of cytoplasmic RNA and DNA sensor pathways. Viral 5′ppp RNA, or longer double-stranded (ds)RNA are recognized by the sensors Retinoic Acid-Inducible Gene I (RIG-I) and Melanoma Differentiation-Associated 5 (MDA5), respectively. Subsequently, activation of these sensors activates Mitochondrial Antiviral Signaling (MAVS) at the mitochondria, which in turn activates two different kinases. Tumor Necrosis Factor (TNF) Receptor Associated Factor 2 (TRAF) Family Member Associated Nuclear Factor κB (NFKB) Activator (TANK) Binding Kinase 1 (TBK1)/Inhibitor of Nuclear Factor Kappa B Kinase Subunit Epsilon (IKKε) phosphorylate Interferon Regulatory Factor 3 (IRF3), which translocates to the nucleus, where it drives type I interferon (IFN) transcription. In addition, the classic IKKα/β/NFκB Essential Modifier (NEMO) kinase complex is activated downstream of TRAF6 (displayed in detail in Figure 2), which results in phosphorylation and degradation of IκBα, the inhibitor of NFκB. Upon release of its inhibitor, NFκB translocates to the nucleus, where it drives transcription of pro-inflammatory response genes. Poly-ubiquitin chains on each molecule are shown in different colors, depending on their linkage type: K11 (light red), K27 (purple), K48 (dark red), K63 (green). Names of deubiquitinating enzymes (DUBs) removing activating K63 poly-ubiquitin chains are listed in green, and K27 chains in purple, whereas E3 ligases adding degradative K11 or K48 chains are displayed in light or dark red.

**Figure 2 viruses-13-00584-f002:**
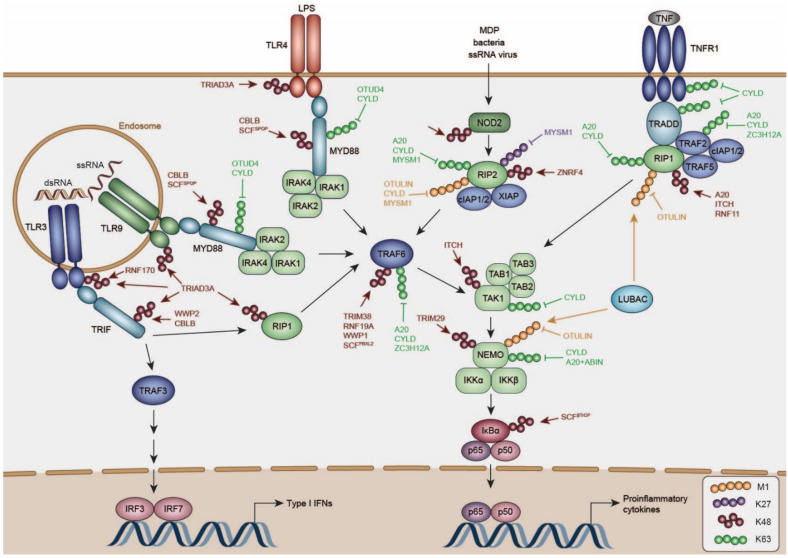
Negative regulation of Toll-like Receptor (TLR), Tumor Necrosis Factor (TNF), and Nucleotide Binding Oligomerization Domain (NOD)-like Receptor (NLR) pathways. (**left**) TLR3 recognizes dsRNA in endosomes, which results in activation of its adapter TRIF. TLR4 is expressed on the plasma membrane and recognizes lipopolysacharide (LPS), whereas TLR9 is expressed in endosomes of specialized myeloid cells, such as plasmacytoid dendritic cells (pDC), and recognizes single-stranded (ss)RNA. Activation of both of these TLRs activates the adapter Myeloid Differentiation 88 (MYD88). (**middle**) NOD2 recognizes bacterial peptidoglycans, thereby activating a complex of cellular Inhibitor of Apoptosis 1 (cIAP1)/cIAP2/X-Linked Inhibitor of Apoptosis (XIAP). (**right**) Binding of TNF trimerizes its receptor, resulting in recruitment of Tumor Necrosis Factor Receptor Type 1 Associated Death Domain (TRADD), Receptor-Interacting Protein 1 (RIP1), TRAF2, TRAF5, cIAP1, cIAP2, and the Linear Ubiquitin Assembly Complex (LUBAC). All these pathways result in the activation of the TGF-Beta Activated Kinase 1 Binding (TAB)/TGF-Beta Activated Kinase (TAK) and classic IKK kinase complexes, which activate NFκB. The IKKα/β/NEMO kinase complex phosphorylates IκBα, the inhibitor of NFκB, subsequently resulting in its degradation, and release of active NFκB. NFκB translocates to the nucleus, where it drives transcription of pro-inflammatory response genes. Poly-ubiquitin chains on each molecule are shown in different colors, depending on their linkage type: M1 (orange), K27 (purple), K48 (dark red), K63 (green). Names of DUBs removing activating K63 poly-ubiquitin chains are listed in green, K27 chains in purple, and M1 chains in orange. E3 ligases adding degradative K48 chains are displayed in dark red.

**Figure 3 viruses-13-00584-f003:**
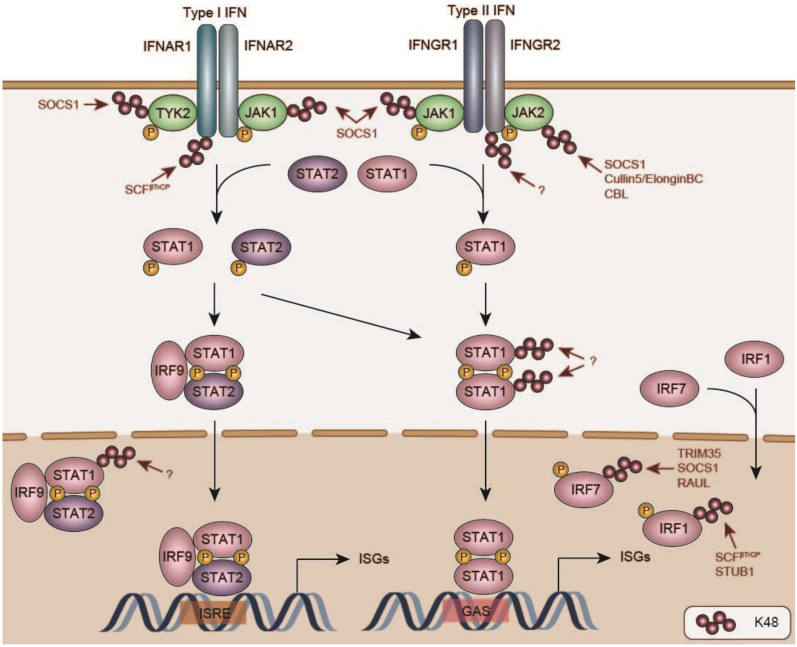
Negative regulation of type I and II IFN signaling. Type I IFNs (IFNα/IFNβ) bind to the IFN Alpha Receptor (IFNAR)1/2 receptor complex. This in turn activates the Janus Kinase 1 (JAK1) and Tyrosine Kinase 2 (TYK2) kinases, which phosphorylate Signal Transducer and Activator Of Transcription 1 (STAT1) and STAT2, resulting in the formation of the heterotrimeric Interferon Stimulated Gene Factor 3 (ISGF3) complex consisting of STAT1/STAT2/IFN-Regulatory Factor 9 (IRF9). Upon translocation to the nucleus, this transcription factor complex drives the expression of several hundred IFN-induced genes (ISG), which harbor an IFN-stimulated Response Element (ISRE) in their promoter regions. In contrast, type II IFN (IFNγ) binds the IFNGR1/2 complex, resulting in JAK1/2 activation, STAT1 phosphorylation, and STAT1 homodimerization. STAT1 homodimers translocate to the nucleus, and associate with Gamma-Associated Sequences (GAS) in the promoters of IFN-Stimulated Genes (ISG). The two illustrative ISGs—IRF1 and IRF7—are shown. E3 ligases adding degradative K48 chains are displayed in dark red.

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
