# Peer review of "Negative Regulation of the Innate Immune Response through Proteasomal Degradation and Deubiquitination"

_viruses, 2021, doi:10.3390/v13040584_

Round 1
Reviewer 1 Report
This is an extremely well-written and comprehensive review article covering how the innate immune response is negatively-regulated by proteasome-mediated degradation and/or the ubiquitin system in general. The review is well-structured and beautifully illustrated - the diagrams, tables and reference collection will therefore be of great interest and use to those established in the field, as well as to newcomers wanting to learn about the complexities of these issues. In this regard, the authors have done a magnificent job in bringing together both older studies and the latest observations, critically reviewing each, as well as setting the scene for what is still not known about this regulation and the future studies that are necessary to dissect new biology. I have no recommendations for alterations to this article and I would be happy to see it published 'as is'.
Author Response
We are very pleased with this positive review, and thank the reviewer for their time and effort. Since this reviewer did not bring up any recommendations for changes to the manuscript, none were made.
Reviewer 2 Report
The review write by Budroni and Versteeg is well wrote and easy to understunding. However, it need of some minor revisions before pubblication.
In particular paragraph 8.
Line 1133-inserte tipe III IFNs. Moreover instert all the type I IFNs.
Line1134 IFNS are not only produce during viral infection (10.1089/jir.2015.0076; 10.1089/jir.2010.0041) This is a rev and this statement should be insert.
Line 1137 and 1138 IFNg is important also in viral infection insert this statement in your paragraph (10.1007/s12250-018-0073-7; https://doi.org/10.3389/fvets.2014.00002; doi/abs/10.3920/978-90-8686-910-7_3). Moreover recent study demonstred that IFNg polarized macrophages to a fenotype more virus-resisten (https://doi.org/10.3390/pathogens9050361).
Author Response
We would like to thank the reviewers for their time and effort reviewing our manuscript, and are pleased with the positive reviews of both reviewers. Please find our responses to their feedback below, as well as a description of the changes made in the revised manuscript. Our replies are in bold.
Reviewer 2:
- Line 1133-inserte tipe III IFNs. Moreover instert all the type I IFNs.
Response to reviewer 2, point 1:
We agree with this reviewer, and inserted type III IFNs, as well as all other type I IFNs in the text here.
Accordingly, line1133 was changed to “Type I (IFNα, and IFNβ), type II (IFNγ) and type III (IFNλ) IFNs are cytokines…”.
Accordingly, lines 1136-1137 were changed to:
“Although other type I IFNs exist with poorly understood functions (IFNδ, IFNε, IFNζ, IFNκ, IFNτ, IFNω), IFNα and IFNβ are particularly recognized for their essential role in antiviral restriction[252].”.
- Line1134 IFNS are not only produce during viral infection (10.1089/jir.2015.0076; 1089/jir.2010.0041)This is a rev and this statement should be insert.
Response to reviewer 2, point 2:
We agree with this reviewer.
Accordingly, lines 1133-1137 were changed to:
“.., cytokines produced during infections, some other cellular stress conditions, and upon aberrant recognition of ‘self’ molecules in auto-immune disorders collectively known as interferonopathies[31,248–250].
- Line 1137 and 1138 IFNg is important also in viral infection insert this statement in your paragraph (10.1007/s12250-018-0073-7; https://doi.org/10.3389/fvets.2014.00002; doi/abs/10.3920/978-90-8686-910-7_3). Moreover recent study demonstred that IFNg polarized macrophages to a fenotype more virus-resisten (https://doi.org/10.3390/pathogens9050361).
Response to reviewer 2, point 3:
We agree with the reviewer on this point, and accordingly added new lines 1141-1143 to reflect the roles of IFNγ in the antiviral response:
“Moreover, IFNγ contributes to the direct antiviral response, as well as indirectly through stimulation of adaptive immunity (reviewed in[257]).”.